# Random subspace-based ensemble classifier for high-dimensional data Using SPARK

**Venkaiah Chowdary Bhimineni**◉, **Rajiv Senapati**◉*

Department of CSE, SRM University, AP, Amaravati, Mangalagiri, Andhra Pradesh, India

◉ These authors contributed equally to this work.
* rajiv.s@srmap.edu.in

## Abstract

High-dimensional data classification remains challenging for machine learning models due to sparsity and overfitting caused by the 'curse of dimensionality'. As the number of features increases, data points become sparse, hindering generalization in classification and leading to higher computational costs and reduced accuracy. To address these issues, we propose an ensemble classifier based on random subspaces implemented in the Spark framework. The proposed framework comprises three key stages. First, the high-dimensional data is normalised through min-max normalisation. Second, the master node partitions the data by using improved deep fuzzy clustering (IDFC). In contrast, the slave node applies support vector machine-modified recursive feature elimination (SVM-MRFE) for efficient feature selection, followed by feature fusion. Finally, we introduced an improved subspace-based ensemble classifier (ISSBEC) that comprises a feature-fusion-based random subspace (FF-RSS), mixed-space enhancement (MSE), and multiple base classifiers. The efficacy of the ISSBEC classifier was evaluated using a set of performance metrics and compared with state-of-the-art methods. Experimental results demonstrate that the proposed approach improves both accuracy and robustness, offering a scalable solution to the limitations of high-dimensional datasets.

## Introduction

High-dimensional data refers to a dataset with a large number of features relative to the number of samples, which can degrade generalization and model performance [1]. High-dimensional data is pertinent to various fields, including biometrics, healthcare, e-commerce, network security, and industrial applications, and requires comprehensive methodologies for managing high-dimensional data. The curse of dimensionality [2], feature identification [3,4], model complexity, and overfitting [5,6] are among the issues in managing high-dimensional data. Several dimensionality reduction techniques and feature selection approaches are proposed, which become the foundation for high-dimensional classification. Feature extraction converts the

**Data availability statement:** https://schlieplab. org/Static/Supplements/CompCancer/datasets. htm.

**Funding:** The author(s) received no specific funding for this work.

**Competing interests:** The authors have declared that no competing interests exist.

original features into a lower-dimensional space, which generates new variables, often using methods such as principal component analysis (PCA) [7], LDA [8], t-SNE [9], LASSO [10,11], feature selection (FS) based on mutual information [12,13], recursive feature elimination (RFE) [14,15] among others.

Classification of high-dimensional data is crucial in numerous real-world applications, including text classification, pattern recognition, and image recognition. However, noise and feature redundancy in such data can severely hinder the effectiveness of the classification approaches. Therefore, it is essential to develop robust feature selection and transformation techniques that generate concise, discriminative, and computationally efficient feature representations. Feature selection strategies for high-dimensional data classification are categorised into filter-based and wrapper-based methods [16]. Wrapper-based approaches achieve better classification accuracy than filter-based approaches. However, these methods are algorithm-specific and computationally expensive [17]. Filter-based methods achieve lower classification accuracy but are computationally less expensive [18]. In wrapper-based methods, the feature space is reduced by eliminating the irrelevant features [19,20]. These methods effectively leverage a feature-relevance strategy to avoid irrelevant searches during the FS process.

To overcome the limitations, researchers have proposed various hybrid feature selection techniques to enhance the performance of high-dimensional data. These include a dual-phase hybrid FS approaches for high-dimensional medical datasets, Maximum Pattern Recognition—Multi-objective Discrete Evolution Strategy (MPR-MDES) [21], a hybrid FS employing a multi-objective genetic algorithm (MOGA) [22], optimized genetic algorithm-based FS [23], adaptive pyramid particle swarm optimisation (APPSO) [24], and binary enhanced golden jackal optimisation (BEGJO) [25], which leverages copula-entropy for increased accuracy. By incorporating filter and wrapper methods, these hybrid techniques improve performance. Still, they require adjusting multiple parameters and have a high computational cost, limiting their applicability to distributed or large-scale settings.

The efficacy of FS approaches is further improved by other methods, such as the dynamic crow search algorithm (DCSA) [26], an adaptive mechanism-based Grey Wolf Optimizer for FS [27], feature filter and group evolution hybrid feature selection (FG-HFS) [28], and a decomposition-based multi-population evolutionary algorithm for feature selection (MPEA-FS) [29]. Random subspace and random projection can improve nearest-neighbour classifier performance in the high-dimensional feature space [30]. It has been observed that data-handling methods remain deficient when utilising FS strategies in existing research. Therefore, further research on the FS approaches for high-dimensional classification is necessary. Machine learning (ML) algorithms, especially ensemble approaches [31], have effectively handled high-dimensional data. Methods such as random forests (RF) [32,33] and gradient boosting machines (GBM) aggregate predictions from multiple models to improve classification accuracy and robustness. These approaches effectively address the challenges of high-dimensional spaces.

Existing approaches face challenges, including class imbalance, redundancy, and computational inefficiency. Therefore, further investigation is needed into the

development of scalable, distributed, and adaptive feature selection models. To address these limitations, this study introduces an ensemble classifier framework based on Apache Spark. This framework integrates an IDFC, an enhanced deep fuzzy classification using an SVM-MRFE approach, and an ISSBEC classifier to facilitate precise, reliable, and scalable classification of high-dimensional data.

The specific contributions of this study are as follows.

- A novel data partitioning technique, IDFC, is implemented in the master node of the Spark framework for efficient high-dimensional data management, which is designed to capture complex, nonlinear relationships between data points and outperform conventional clustering methods when applied to high-dimensional feature space.

- We propose the SVM-MRFE method at the slave node of the Spark framework for feature selection to minimize dimensionality in high-dimensional data. It removes less significant features while preserving the most important features for classification based on their rank.

- We propose the ISSBEC architecture for classifying high-dimensional data, which is organized into $n$ number of blocks, each containing three essential components: FF-RSS, MSE, and a base classifier. It effectively classifies high-dimensional data and enhances data representation through a multi-block structure. It integrates various base classifiers for efficient processing to improve accuracy and robustness.

- We proposed an improved k-nearest neighbour (IKNN) as one of the base classifiers for enhanced generalization and reduced overfitting, achieving more accurate predictions in high-dimensional data classification.

- We conducted comparison experiments across various high-dimensional binary-class datasets to demonstrate the superiority of our method over other methods.

The remainder of the paper is organized as follows. Literature Review Section provides a summary of the state-of-the-art literature related to conventional high-dimensional data classification methods. Proposed Methodology Section provides a detailed implementation of the proposed ISSBEC architecture for high-dimensional data classification using Apache Spark. Result Section discusses experimental findings and highlights the efficacy of the proposed ISSBEC approach. Finally, Conclusion Section concludes the paper with a summary and review of the key findings.

## Literature review

This section reviews existing research on high-dimensional classification and delivers recent enhancements in ML-based approaches. These approaches play a significant role in high-dimensional classification, as they enhance overall data analysis, improving the accuracy, and offer practical data-handling methods. By integrating advanced FS and dimensionality reduction techniques, ML approaches overcome various challenges while performing high-dimensional classification.

Several recent approaches have explored enhancements in high-dimensional data classification. For example, the sparse kernel K-means (SKKM) approach [34], which enhances clustering performance for high-dimensional data by selecting relevant features while penalizing redundant ones. Likewise, a depth-based nearest-neighbour approach [35] for effective high-dimensional classification, overcoming the limitations of the traditional k-nearest-neighbour method, by carefully identifying low-dimensional features in high-dimensional data using information-gain-based subspace clustering (IGSC). Furthermore, the enriched RF (ERF) [36] is used for weighted random sampling to prioritise informative features, thereby improving accuracy and computational efficiency, and leave-one-out cross-validation (LOOCV) is used to reduce complexity while maintaining accuracy. To balance computational speed and accuracy in feature selection, hybrid feature selection frameworks have also been examined. Hybrid Dimensionality Reduction Forest with Pruning Framework(HDRFPF) [37] for high-dimensional data classification, addressing issues such as information loss, noise, and redundant feature vulnerability, by incorporating bagging with tree-based FS for efficient feature splitting and diversity

with training subsets. The hybrid feature selection algorithm [38] is based on improved interaction information, and a multitasking-based particle swarm optimisation (PSO) approach [39] was introduced to strengthen the feature relevance for high-dimensional data classification. These hybrid and evolutionary methods demonstrate that mutual information, clustering, and adaptive weighting can improve model discrimination in high-dimensional settings. Metaheuristic algorithms have effectively handled feature selection for high-dimensional data. For instance, the Dynamic Crow Search Algorithm (DCSA) [37] was proposed to improve the categorisation accuracy of high-dimensional biomedical data. The Feature filter and group evolution hybrid feature selection (FG-HFS) [28] uses spectral clustering to place features into groups based on their relationships, and symmetric uncertainty to eliminate features that do not belong to these groups. Similarly, a High-Dimensional Ensemble Learning Classification (HDELC) algorithm [31] produced a feature space reconstruction matrix that optimizes feature selection and reconstruction for high-dimensional data. This optimal feature space improves the representativeness of the ensemble model.

Recent research has also examined ensemble and semi-supervised frameworks for high-dimensional classification. The feature selection-based semi-supervised classifier ensemble (FSCE) [1] and adaptive semi-supervised classifier ensemble (ASCE) frameworks enhance the adaptability of weighted classifier ensembles. The semi-supervised random subspace classifier ensemble (SSRS) and adaptive semi-supervised random subspace classifier ensemble (ASSRS) approaches [40] are used to reduce the data and feature dimensions of complex datasets, identify subspaces, label samples, and assign classifier weights, thereby minimizing sample size for data-driven predictors. Meanwhile, Envelope rotation forest [41] for inadequate separability, limited diversity, and increased sensitivity, adaptive classifier ensemble learning method (AdaSPEL) [42] for local space perceptron and, Classifier Ensemble Method Based on Subspace Enhancement (CESE) [43] features a sophisticated SSE for efficient feature selection and transformation, creating various effective feature subspaces. It also incorporates an MSE to develop diverse feature representations through multiscale rotation and fusion. The random subspace and random projection techniques for ensembling nearest-neighbour classifiers [30] in the high-dimensional feature space were compared with the traditional nearest-neighbour approaches, and the methods were tested on microarray, image, and chemoinformatics data. The random projection method performs significantly better than the random subspaces for most datasets. However, these are limited to dual classification and require improvement for multi-class problems. Recent advancements have been achieved in adaptive subspace clustering and Spark-based ensemble systems. A novel adaptive multi-view subspace clustering method [44] addresses the challenges of using high-dimensional multi-view data, specifically the presence of irrelevant features, and assigns weights to data views based on the compactness of the clusters. Recent studies on multimodal learning [45] focus on integrating heterogeneous data to improve robustness. Artificial intelligence and multimodal learning analytics showed that feature fusion enhances the interpretation of challenging data.

Randomized optimization approaches [46] limit the number of variables to random subspaces by employing various data-adaptation policies. Recently, it has been demonstrated that combining SPARK, ML, and DL [47–49] provides a feasible solution for accurate, scalable classification. Based on deep subspace sequential clustering, a new online anomaly detection model, NADHS [50], was introduced for high-dimensional real-time data. For large-scale data clustering, a distributed subspace ensemble approach called the subspace cluster ensemble (SSCE) [51], which employs random partitioning and ball fusion, was introduced. To mitigate class imbalance, an enhanced broad learning system with adaptive locality preservation (IBLS-ALP) [52], an incremental adaptive subspace ensemble designed to preserve the local characteristics of small-sample datasets. However, ADMTSK [53], a fuzzy system that is an adaptive TSK and uses the Dombi T-norm to reason in high dimensions. A multilayer jointly The evolving and compressing fuzzy neural network (MECFNN) [54] is a multilayer fuzzy ensemble with self-adaptive compression, developed for high-dimensional classification. To learn self-representation matrices for multi-view data end-to-end, the Multi-view Deep Subspace Clustering Network (MvDSCN) [55] uses a dual network structure made up of a diversity network (Dnet) and a universality network (Unet). Deep convolutional autoencoders are used to create a

latent space in which Unet finds a common matrix that applies to all views, while Dnet concentrates on view-specific matrices. The model successfully captures nonlinear and high-order relationships between various perspectives by using the Hilbert-Schmidt independence criterion (HSIC) as a diversity regularizer. The authors of [56] created an effective, scalable feature extraction technique based on Apache Spark that quickly and efficiently extracted highly significant features from millions of genome sequences. Five stages of feature extraction were carried out by their method: sequence length, nucleotide base frequency, nucleotide base pattern organisation, nucleotide base distribution, and sequence entropy. This procedure resulted in a 14-dimensional fixed-length numeric vector that allowed for the unique representation of every genome sequence. Medical prognostication is also performed using deep learning ensemble methods. Nested Ensemble Deep Learning for Gynaecological Cancer Prediction (NEDL-GCP) [57] is predicting the risk of cancer. In contrast to their deep framework, our ISSBEC method uses distributed, high-dimensional data, scaled using Spark, for ensemble classification. The comparison of FF-RSS and MSE with the recent approaches is demonstrated in Table 1. All of these findings suggest that current research on subspace and ensemble approaches is encouraging, but has significant limits due to data imbalance, repetition, and inadequate distributed system optimisation. The current study addresses this challenge using Spark, which combines clustering, feature selection, and ensemble learning into a single framework. The proposed Spark-based ISSBEC classifier addresses the above limitations. In the proposed system, Spark employs effective data partitioning using IDFC and FS with SVM-MRFE, and ISSBEC employs FF-RSS for subspace generation, MSE, and various base classifiers for high-dimensional classification. A summary of the notations employed in this study is provided in Table 2.

## Proposed methodology

Handling high-dimensional data is significantly challenging due to the "curse of dimensionality," which exponentially increases the volume of feature space, making it difficult to analyze. These challenges may lead to model overfitting, decreased efficacy, and increased computational cost. To address the challenges of high-dimensional data classification, this study proposed an ISSBEC framework leveraging the distributed computing capabilities of Spark, which enables efficient processing of high-dimensional datasets, which is illustrated in Fig 1. The proposed framework consists of three stages, namely 1) Pre-processing, 2) Spark framework, and 3) Classification. All the stages are structured to facilitate distributed, parallel, and scalable processing of high-dimensional data.

### Preprocessing by Min-Max normalization

This phase ensures that raw data is useful for analysis by enhancing data quality and improving model performance, thereby ensuring reliable results. Effective preprocessing is critical for high-dimensional data due to the difficulties of handling large feature sets. Min-max normalization is a key preprocessing technique that addresses issues arising from differing scales and ranges across features in high-dimensional datasets. This method rescales the feature values to a standard range, typically [0, 1], ensuring consistency and comparability across features. The process involves

**Table 1. Comparison of FF-RSS＋MSE with recent approaches.**

| Reference | Framework | Primary Focus | Approach | Fusion/Adaptability |
|-----------|-----------|---------------|----------|---------------------|
| [50], 2025 | Ensemble | HD anomaly detection | Deep Subspace Clustering | Partial |
| [51], 2025 | Distributed | HD clustering | Random Subspace Fusion | Partial |
| [52], 2025 | Incremental | Imbalanced HD data | Adaptive Subspace Ensemble | Yes |
| [54], 2024 | Fuzzy Network | HD classification | Multilayer Compression | Yes |
| [53], 2025 | Adaptive Fuzzy | HD reasoning | Adaptive TSK Ensemble | Yes |
| Proposed Plan | Spark | HD classification | Hybrid Random Subspace＋Mixed Space Rotation | Fusion＋Adaptive Balance |

**Table 2. Notations used in this work.**

| Symbol | Description |
| --- | --- |
| $\beta$ | Fuzzy clustering centers |
| $X_m$ | Hidden features of $m^{th}$ batch |
| $\beta_m$ | Pseudo labels |
| $r_m$ | Fuzzy memberships for the data in $m^{th}$ batch |
| Aff | Affinities between data points |
| B | Biases |
| W | Weights |
| $i^{Norm}$ | Normalized data |
| $\Re^{\nu}$ | Hidden feature vector |
| $\Upsilon_{x,y}$ | Fuzzy membership for each data point |
| $\mu$ | Mean of the distance between clusters |
| $T_m$ | Target for the $m^{th}$ batch |
| $\omega$ | Fixed kernel function in DFC |
| $\eta_1, \eta_2$ | Hyperparameters in DFC |
| $\delta$ | Affinity hyperparameter |
| $\omega^{Hybrid}$ | Hybrid kernel function |
| $k^{Gaussian}$ | Gaussian kernel function |
| $k^{Exponential}$ | Exponential kernel |
| $J, J^{'}$ | Input space vectors |
| $P^{i^{Norm}}$ | Partitioned data generated by IDFC |
| $\mu_d^a$ | Total mean of the $b^{th}$ feature. |
| $\mu_d$ | Mean vector |
| f | feature of partitioned data |
| V | total no.of classes |
| $\mu_0$ | Mean of partitioned data |
| $S^f$ | Selected Features |
| $\varepsilon$ | Mixed enhanced features |
| $F^f$ | Fused feature set |
| $\vartheta$ | Set of Subspace enhanced feature |
| n | New instance |
| $\bar{n}$ | All training instances |
| $E_{dis}$ | Euclidian distance |
| $WM_{dis}$ | Weighted Minkowski distance |
| $z_x, z_u$ | Each datapoint's fuzzy membership |
| $\| \bullet \|$ | Euclidian |
| $i_q^{Norm}$ | Normalized scores |
| $s(f^a)$ | Fisher score of $a^{th}$ feature |
| $Ms(f^a)$ | Modified fisher score of $a^{th}$ feature |

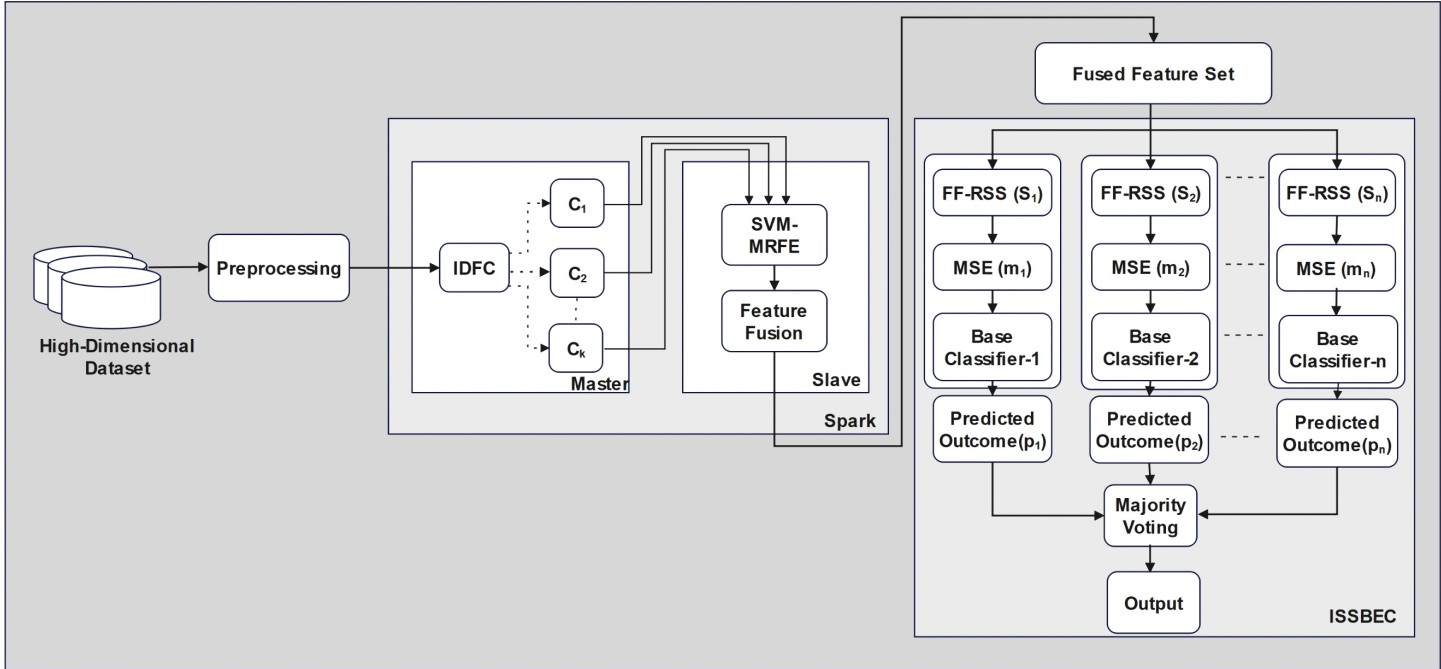

**Fig 1. Proposed architecture for High-dimensional data classification.**

transforming a vector of scores $i = (i_1, i_2, i_3, \ldots, i_q, \ldots, i_n)$, where $i_q$ represents the index q score and $n$ signifies the overall number of scores, into normalized scores $i^{Norm}$ using Eq 1 [58].

$$i^{Norm} = \frac{i - min(i)}{max(i) - min(i)}$$

(1)

where *max(i) and min(i)* denote the maximum and minimum feature values of the raw scores in the dataset, respectively. This process preserves the original distribution of the data while transforming all features to a common scale, effectively preparing high-dimensional data for classification.

## Spark framework processing

The Spark framework [59] is used for robust distributed computing and efficient in-memory data processing. It contains two main components, namely 1) the master node, which is responsible for data partitioning using the IDFC, and 2) the slave node is responsible for FS by using the SVM-MRFE method, a modified version of the conventional SVM-RFE approach, which systematically removes less important features while retaining the most contributing features for the classification task, then followed by feature fusion, which is illustrated in Fig 1 and the Algorithm 1 depicts the entire Spark framework processss.

**Data partitioning in master node.** In the Spark framework, the master node is the central coordinator for executing the Spark distributed process, resource allocation, and ensuring seamless collaboration among various nodes. In this phase, the master node used IDFC to partition the normalized data $i^{Norm}$ and it enhances the data partitioning by integrating a hybrid kernel function that blends Gaussian and exponential kernels, allowing it to effectively capture complex nonlinear manifold structures within high-dimensional data. This hybrid formulation enriches the similarity

representation between samples, enabling the clustering mechanism to model both local smoothness and global variations more accurately.

The parameters for this process include $C_i$, which denotes the number of clusters, and $b_n$, which denotes the batch size. In this case, the number of clusters was set to three, denoted as $C_1$, $C_2$ and $C_3$, and each cluster contained data points. The partitioning of input normalized data ($P^{i^{Norm}}$) and the partitioned batches is specified as $b/b_n$ such that each batch is denoted by $i^{Norm_m}$, where $m = 1,2,3,...,b/b_n$.

A deep neural network can essentially map the data into any desired distribution. Overfitting can be avoided by using an autoencoder in deep clustering methods [60,61]. The autoencoder recovers the original data by decoding them with another neural network [62].

### Algorithm 1. Spark Framework Process

```
Input: Normalized dataset (i^Norm)
Output: Fused feature set (F^f)
Master Node:
1: Partition the data using IDFC.
2: The partitioned data P^{i^{Norm}} is sent to the slave node for further processing.
   Slave Node:
3: For each partition P^{i^{Norm}}:
   • Feature selection is performed by using SVM-MRFE.
   • Store the selected features (S^f).
4: End For
5: Generate fused features F^f from the selected features (S^f) from Step 3.
```

The loss function for the autoencoder is expressed in Eq 2, where $\|.\|$ indicate the Euclidean norm, $Z(W)$ represents a regularization term, and $R_{W,B}(i^{Norm_x})$ specifies the reconstruction term.

$$L(i^{Norm}, \Theta) = \frac{1}{b} \sum_{x=1}^{b} \|R_{W,B}(i^{Norm_x}) - i^{Norm_x}\|^2 + \gamma.Z(W)$$

(2)

Let us assume that the set of hidden representations be $X = (z_1, z_2, z_3, ...., z_x, ..., z_b)$, where each $z_x \in \Re^\nu$ represents a hidden feature vector. The fuzzy membership $\Upsilon_{x,y}$ for each data point to the cluster is computed using Eq 3. In Eq 3, $fuz$ is the fuzzifier parameter, and $\mu$ balances the distance within and between clusters.

$$\Upsilon_{x,y} = \frac{[\|z_x - \beta_y\|^2 - \frac{\mu}{\sum_{v=1}^{\nu} \|\beta_v - \overline{\beta}\|^2} \|\beta_y - \overline{\beta}\|^2]^{\frac{-1}{(fuz-1)}}}{\sum_{v=1}^{\nu} [\|z_x - \beta_v\|^2 - \frac{\mu}{\sum_{v=1}^{\nu} \|\beta_v - \overline{\beta}\|^2} \|\beta_y - \overline{\beta}\|^2]^{\frac{-1}{(fuz-1)}}}$$

(3)

Pseudo-labels $\beta_m$ are extracted from fuzzy memberships $r_m$, and the target $T_m$ for the $m^{th}$ batch is expressed in Eq 4.

$$T_m = \frac{\frac{\Upsilon_{x,y}^2}{\sum \Upsilon_{x,y}}}{\sum_{v=1}^{\nu} [\frac{\Upsilon_{x,y}^2}{\sum_x \Upsilon_{x,y}}]}; \sum_{y=1}^{\nu} T_{x,y} = 1 \forall y$$

(4)

The Kullback-Leibler (KL) divergence loss function is mathematically formulated in Eq 5. We minimize this loss function to enhance the accuracy of the fuzzy membership predictions.

$$minKL(T\|K) = min \sum_{x=1}^{b} \sum_{y=1}^{\nu} T_{x,y} log \frac{T_{x,y}}{\Upsilon_{x,y}}$$

(5)

The Eq 6, $Aff_{x,u}$ represents the affinity between data points $i^{Norm_x}$ and $i^{Norm_u}$, with a higher affinity leading to smaller distances in the feature space. The expression for the loss function is formulated in Eq 7.

$$minW_z = min \sum_{x,u=1}^{b} \|z_x - z_u\|^2 Aff_{x,u}$$

(6)

$$M_L = \sum_{x=1}^{b} \|R_{W,B}(i^{Norm_x}) - i^{Norm_x}\|^2 + \eta_1 \sum_{x=1}^{b} \sum_{y=1}^{\nu} T_{x,y} log \frac{T_{x,y}}{\Upsilon_{x,y}} + \eta_2 \sum_{x,u=1}^{b} \|z_x - z_u\|^2 Aff_{x,u}$$

(7)

$$Aff_{x,u} = \begin{cases} \frac{exp[-\frac{\|z_x-z_u\|^2}{\omega}]}{\delta} & ; \quad \ell_x = \ell_u \\ 0 & ; \quad \ell_x \neq \ell_u \end{cases}$$

(8)

where the pseudo label is denoted as $\ell_x$, the fixed kernel function in deep fuzzy clustering(DFC) method is indicated as ω, the hyperparameters are specified as $\eta_1$ and $\eta_2$, the affinity hyperparameter, which controls the scale of the affinity, is indicated as δ, and the label is specified as $\ell_u$.

Traditional DFC [59,62] has a single fixed kernel, rendering it incapable of capturing non-linear features in high-dimensional spaces. The IDFC technique exploits the reproducing kernel Hilbert space (RKHS) property by combining Gaussian and exponential kernels (hybrid kernel), enabling complex, nonlinear similarities in the input space to be represented as linear features in a higher-dimensional feature space.

The formulation of the proposed hybrid kernel function is expressed in Eq 9.

$$\omega^{Hybrid} = \alpha * k^{Gaussian}(J, J') + (1 - \alpha) * k^{Exponential}(J, J')$$

(9)

In Eq 9, the Gaussian kernel [63] is indicated as $k^{Gaussian}(J, J')$, which is an example of a radial basis, and the exponential kernel is specified as $k^{Exponential}(J, J')$, which is closely related to the Gaussian kernel.

$$k^{Gaussian}(J, J') = [-\frac{\|J - J'\|^2}{2\sigma^2}]$$

(10)

$$k^{Exponential}(J, J') = exp[-\frac{\|J - J'\|^2}{\sigma}]$$

(11)

In Eqs 10 and 11, $\|.\|$ indicates the Euclidean norm, σ is a parameter that controls the kernel's width, J and $J'$ are vectors in the input space.

The proposed IDFC convergence aligns with the properties of the fuzzy C-means (FCM) objective function, which is bounded by zero and monotonically reduced with each step. Since the hybrid kernel is positive semi-definite, adjusting the fuzzy membership $\Upsilon_{x,y}$ and cluster centres $\beta_y$ always results in an objective function $J_m = \sum_{x=1}^{n} \sum_{y=1}^{c} \Upsilon_{x,y}^m \|z_x - \beta_y\|^2$ that does not decrease. It can be confirmed that IDFC guarantees convergence to any local optimum point of $J_m$ while adhering to the constraint $\sum_{y=1}^{c} \Upsilon_{x,y} = 1$. In high sparsity, the hybrid Gaussian-Exponential kernel is employed to avoid the instability of a regular fuzzy C-means in high-dimensional feature spaces. It avoids violating the Mercer condition and stabilises membership updates by smoothing sparse similarities. This analysis confirms that IDFC shares the convergence properties of fuzzy C-means, and exhibits enhanced optimality with sparse data due to the adaptive weighting factor in the kernel α.

The IDFC-generated partitioned data is denoted as $P^{i^{Norm}}$, with each partition being processed by a slave node. Consequently, the hybrid kernel function enhances the effectiveness of the fuzzy clustering process while reducing computational complexity compared with the traditional DFC method. This advancement in the IDFC method makes it more efficient than the conventional DFC method for data partitioning.

## Feature processing in slave node

The slave node is responsible for feature selection on the partitioned data $P^{i^{Norm}}$ and for feature fusion within each partition. The FS is performed using the SVM-MRFE method, which enhances the conventional SVM-RFE. After the FS process, the remaining features are fused within the slave node. This process combines features to generate a new set that enhances the classifier's ability to differentiate between classes by leveraging multiple feature sets. The feature selection and feature fusion processes are as follows.

**Feature selection by modified recursive feature elimination**: In the Spark framework, FS plays a vital role in high-dimensional classification. Conventional SVM-RFE [64] has been widely used for FS. However, they have certain limitations, especially when dealing with datasets containing many correlated features. To address this issue, SVM-MRFE approach is proposed in this research. SVM-MRFE uses the discriminative power of an SVM to evaluate the importance of features based on their contribution to the decision boundary. Unlike traditional RFE, which may struggle with noisy or redundant features, SVM-MRFE accurately identify the most informative features and results in the removal of less relevant attributes while retaining those that maximally enhance class separability. The final selected feature set improves classification accuracy, reduces overfitting, and lowers computational cost by focusing the model on the most discriminative dimensions of the data.

The conventional SVM-RFE method follows a straightforward sequence of steps for FS:

- Training the SVM Classifier: The SVM model is trained on the partitioned data ($P^{i^{Norm}}$) obtained from the master node to create a classification model.

- The Ranking Criterion for all Features: The computed significance of each feature was evaluated based on the weights or support vectors generated by the SVM classifier.

- Remove Features with the Smallest Ranking Values: Features that contribute to the model's performance are removed iteratively, focusing on those with low ranking scores.

However, SVM-RFE may not be the most effective method for datasets with strong feature correlations and can be computationally expensive when applied to high-dimensional data. To address the above issues, hybrid feature selection approach SVM-MRFE was introduced as an enhanced FS approach, which outperforms conventional SVM-RFE, and relies solely on margin-based ranking and can be influenced by correlated features and local optima. This method integrates the Modified Fisher Score with SVM-RFE to balance feature evaluation by accounting for both their statistical separability and the classifier's contribution. By using the Eq 12, the hybrid ranking score of each feature $f_i$ was determined.

$$R(f_i) = \alpha \cdot \frac{|w_i|^2}{\max(|w|^2)} + (1 - \alpha) \cdot \frac{M_s(f_i)}{\max(M_s(f))} \tag{12}$$

where $\alpha \in [0, 1]$ controls the contribution balance (set to 0.5 in this study), $Ms(f_i)$ is the modified Fisher score, and $W_i$ is the SVM weight for feature $f_i$. Using backward elimination, $R(f_i)$ ranks and eliminates low-scoring features. This hybridisation combines SVM optimisation margins and Fisher-type separability, ensuring stability, generalisation, and local optimum reduction in high-dimensional data. Algorithm 1 describes the synchronization process between the master and slave nodes and illustrates the overall data flow of the hybrid feature selection and feature fusion within the Spark system.

 

The optimised Fisher score used in SVM-MRFE is derived from established principles of discriminant analysis, where the ratio of between-class to within-class variance quantifies the discriminatory efficacy of a given feature. By combining this score with SVM's margin-based weights, SVM-MRFE creates a criterion that maximises both hyperplane margin and statistical separability. The proposed SVM-MRFE method improves on the conventional SVM-RFE method through the following steps:

- Train the SVM Classifier: Similar to SVM-RFE, the SVM model is initially trained on the partitioned data ($P^{i^{Norm}}$) obtained from the master node.

- The conventional Fisher score [65] and modified Fisher score expressions are represented in Eq 13 and Eq 14.

$$s(f^a) = \frac{\sum_{d=1}^{N} \mu_d(\mu_d^a - \mu^a)^2}{(\sigma^a)^2} \; where, \; (\sigma^a)^2 = \sum_{d=1}^{N} \mu_d(\sigma_d^a)^2 \tag{13}$$

$$Ms(f^a) = \frac{\sum_{d=1}^{N} \mu_d(\mu_d^a - \phi(\bar{f}))^2}{(\sigma^a)^2} \; where, \; (\sigma^a)^2 = \sum_{d=1}^{N} \mu_d(SE(\bar{f})) \tag{14}$$

$$\mu_d^a = \frac{1}{V} \sum_{d=1}^{V} f_d \tag{15}$$

$$\phi(\bar{f}) = \frac{\bar{f} - \mu_0}{SE(\bar{f})} \tag{16}$$

In the above equations, $\mu_d^a$ indicates the local mean of $a^{th}$ the feature, $\mu_d$ denotes the mean vector, $\mu^a$ and $\sigma^a$ denotes the mean and standard deviation values of the $a^{th}$ feature across the entire dataset; SE($\bar{f}$) specifies the whole data standard error and $\phi(\bar{f})$ indicates the whole data sample mean. The feature of partitioned data $P^{i^{Norm}}$ is indicated as f, the total number of classes is denoted as V, and the partitioned data mean is denoted as $\mu_0$.

A modified version of Fisher ranking, defined by a convex, monotonically decreasing curve related to within-class variance, is used in the SVM-MRFE procedure to remove the least important features. Every iteration uses a convex quadratic algorithm to recalculate the SVM margin, ensuring that the optimization is always within a convex feasible region. This implies that it will eventually converge to a subset of features. Compared with traditional RFE, which can oscillate due to substantial differences among highly correlated features, SVM-MRFE will provide better convergence because it employs the modified Fisher score's separability criterion, which introduces a strong convexity component. To avoid overfitting in high-sparsity, one can take advantage of the Fisher-denominator regularity, and the solution with the final subset is comparable to the one with minimal empirical risk. Therefore, this process attains asymptotic optimality in maximising the margin.

- Evaluation of the ranking criteria for all features: Using the modified Fisher score, each feature is ranked according to its relevance and contribution to the classification task.

- Remove Features with the Smallest Ranking Values: Features with the lowest ranking values are eliminated, as in the SVM-RFE process, but with improved criteria derived from the Modified Fisher score.

This enhanced method highlights the most significant features and simplifies the model, improving its interoperability and efficacy. Consequently, the SVM-MRFE method optimizes FS for HD data, retaining only the most relevant features

$S^f$ for the subsequent classification stage, thereby improving both performance and computational efficiency compared to the conventional SVM-RFE method. The selected features ($S^f$) from different partitions are fused to form the fused(combined) feature set $F^f$. Feature fusion is a method used to combine various feature sets to create a more informative and comprehensive representation of data for ML tasks. Although feature selection reduces dimensionality by retaining only the most informative features, redundancy may still exist among the selected features. Feature fusion combines these selected features into a compact form, reducing redundancy and improving the stability and robustness of the classifier. This step can enhance generalization by capturing the most discriminative information in a smaller, fused feature space, leading to improved classification performance while maintaining computational efficiency. By fusing only the relevant features, the process avoids unnecessary information loss that would occur if fusion were applied to the entire high-dimensional feature set.

**High-dimensional data classification by ISSBEC architecture.** The final classification phase is performed using the ISSBEC architecture, which employs a multi-block structure to improve performance via diverse classifiers and feature-space optimization methods. Each block contains FF-RSS, MSE, and a base classifier as illustrated in Fig 1. Algorithm 2 depicts the entire ISSBEC process and the description of each component as follows:

**Algorithm 2. ISSBEC Process**

```
   Input: Fused feature set(F^f)
   Output: Output(O_ISSBEC)
1: for i in1, 2,..., n
2: Generate the number of subspaces(S_i) from the fused feature set(F^f).
3: Select the number of features(M) for each rotation subspace.
4: Randomly divide the subspace(S_i) features into 'l' disjoint subspaces DSS_i,j(j=1,2,—,l);
5: for j in 1,2,....,l
   • Construct subspace dataset X_i,j from S_i for training.
   • Select 75% samples from X_i,j using bootstrapping to get a new training setX'_i,j;
   • apply PCA on X'_i,j to obtain C_i,j coefficients. i.e., C_i,j = PCA.fit(X'_i,j)
6: end for
7: Design rotation matrix R_i from C_i,j
8: Concerning the original features rearrange R_i to R_i^a
9: Retrieve new train set X_final^i using R_i^a
10: Apply base classifier on every X_final^i
11: end for all subspaces
12: Use majority voting for an ensemble of all the base classifier predictions
```

**Feature Fusion based Random Subspace (FF-RSS).** Random Subspace [66] is an ensemble learning technique that creates various classifiers by selecting random subspaces from the feature space. This method enhances the model's overall performance by increasing the diversity of classifiers constructed exclusively from the features in their respective subspaces. The FF-RSS operates on a fused feature space, which is generated from each Spark partition by integrating distributed feature sets into a single feature space, which contains inter and intra-partition feature information. By using this feature space, FF-RSS generates random subspaces for each learner. For every subspace $S_i$ is calculated by using the Eq 17.

$$S_i = F_i W_i + \lambda(F_i \cap F_j) \tag{17}$$

Here, $W_i$ is denoted as a random weight vector, $F_j$ represants randomly choosen sub-feature set, $\lambda$ represents the control degree of overlapping between different feature subspaces. This design provides diversity over the subspaces, with the help of random subspace selection and uses global information in the dataspace.

**Mixed space enhancement.** MSE [43] is an advanced feature enhancement method that combines multi-scale rotation reconstruction with subspace features to produce a diverse and robust set of features. By randomly selecting

the number of features in each rotation subspace across different scales for each base classifier, the ensemble system's diversity increased by MSE. This stage utilises multi-scale rotation reconstruction to produce diverse mixed-enhanced features. Inorder to enhance the diversity, MSE refines the random subspaces by gathering orthogonal transformed versions of its elements. The MSE defines $H_i$ as the enhanced version of random subspace, which is reprasented in Eq 18.

$$H_i = \alpha S_i + (1 - \alpha)R(S_i) \tag{18}$$

Here, $R(.)$ repraents the random rotation transformation and $\alpha \in [0, 1]$ is the mixing coefficient. This process balancing the ability to preserve the essential features with improved diversity by using random variations.

The subspace features obtained using FF-RSS offer a more compact and effective representation than the original feature space. This multi-scale feature fusion mechanism enhances feature expression and ensures high scalability, as rotation-based feature fusion can be replaced with other operations if needed. Thus, the mixed enhanced features obtained from the MSE components $(m_1, m_2, ---, m_n)$ in each block are specified as $\varepsilon_1, \varepsilon_2, ----.\varepsilon_n$.

**Importance of base classifiers.** The base classifier is an ML algorithm that can produce predictions independently. That is the foundation for complex classification strategies, particularly ensemble learning. In the proposed ISSBEC classifier, the mixed enhanced features $(\varepsilon_1, \varepsilon_2, ---, .\varepsilon_n)$ generated from MSE components $(m_1, m_2, ---, m_n)$ for blocks $Block-1, Block-2, ---, Block-n$ are fed into the base classifiers of respective blocks to produce predictions. We used three base classifiers in the proposed ISSBEC classifier to analyze the model; these are discussed below.

**Random Forest (RF)**: The RF classifier [67,68] is a highly effective supervised learning model used for both regression and classification tasks. For high-dimensional data, overfitting was reduced by combining multiple trees' results. It uses ensemble learning, combining predictions from multiple decision trees (DTs) to improve overall model performance. Over-all, the DT classifier uses a diverse ensemble of DT to balance bias and variance, resulting in an ISSBEC architecture for high-dimensional classification. The predicted outcome is obtained by mixed enhanced features from MSE.

**Support Vector Machine (SVM)**: The SVM [69] for high-dimensional classification is an effective two-class classifier that aims to construct an optimal separating hyperplane maximising the margin between classes. SVMs are effective at handling sparse data and can identify complex, non-linear decision boundaries, making them particularly suitable for high-dimensional datasets. In the wrapper framework, SVM evaluates subsets of features based on their predictive per-formance, ensuring that the selected features are directly optimized for classification. Although they can encode complex relationships, nonlinear kernels and metaheuristic feature selectors significantly raise processing costs and decrease interpretability. SVMs can handle both linear and non-linear decision boundaries, with the Gaussian *rbf* kernel used for non-linear cases to map features into a high-dimensional space where linear separation is possible. Leave-one-out cross-validation method was used to assess the performance of the SVM model and estimate the misclassification rate. This approach involves removing a single sample from the training set, training the model on the remaining data, and testing the excluded sample on the resulting hyperplane. By repeating this process for each sample, the total number of misclassifications was used to estimate the model's risk. In summary, the SVM classifier leverages mixed enhanced features from MSE to build an optimal hyperplane for class separation and uses cross-validation to assess model perfor-mance, resulting in the predicted outcome.

**Improved K-Nearest Neighbours (IKNN)**: In the ISSBEC architecture, the IKNN classifier is a critical component that effectively classifies high-dimensional data. It makes predictions based on a combination of enhanced features from MSE. This advanced classifier extends the conventional KNN algorithm [70] by integrating several enhancements that signifi-cantly improve its performance and generalization capabilities. The basic steps of the KNN algorithm involve computing the distances between a new instance $(n)$ and all training instances $(\bar{n})$ using a specified distance metric. The Euclidean distance is commonly used for this calculation, mathematically formulated in Eq 19.

$$E_{dis}(n, \overline{n}) = \sqrt{\sum_{i=1}^{t} (n_i, \overline{n_i})}$$

(19)

However, Euclidean distance has limitations, such as treating all features equally essential and being sensitive to feature scaling. The IKNN classifier uses Weighted Minkowski Distance to address these issues. Unlike Euclidean distance, which does not account for the varying significance of different features, Weighted Minkowski Distance allows assigning different weights to features based on their relevance. This approach enables the IKNN classifier to better generalize from the training data to new, unseen instances, mitigate overfitting by considering feature weights, and achieve more accurate classifications by reflecting the true importance of the features in the distance metric. The mathematical formulation of the Weighted Minkowski Distance is expressed in Eq 20.

$$WM_{dis}(n, \overline{n}) = [\sum_{i=1}^{t} W_i(n_i, \overline{n_i})^\rho]^{\frac{1}{\rho}} ; \rho \geq 1$$

(20)

where $W_i$ is the weight and is determined using Eq 21.

$$W_i = \frac{\nu_{p_i}}{\sum_{i=1}^{t} \nu_{p_i}}$$

(21)

$\nu_p$ denotes the value parameter and is determined using Eq 22

$$\nu_{p_i} = 1 - (acc_0 - acc_e)$$

(22)

## Algorithm 3. IKNN algorithm

```
    Input: Features from MSE (m_i)
    Output: Class of sample (m_i)
1: Find the value parameter by using Eq 22,
    i.e., ν_{p_i} = 1 - (acc_0 - acc_e)
2: Find the weight of each feature, using Eq 21,
    i.e., W_i = ν_{p_i} / Σ_{i=1}^{t} ν_{p_i}
3: Evaluate the Weighted Minkowski distance using Eq 20,
    i.e., WM_dis(n, n̄) = [Σ_{i=1}^{t} W_i(n_i, n̄_i)^ρ]^{1/ρ}; ρ ≥ 1
4: Find the average distance of each class [1/n Σ WM_dis(n, n̄)]
5: Select the nearest neighbour based on the average distance
6: Return predicted outcome.
```

Here, $acc_e$ indicates the average accuracy of the conventional KNN (k = 3,5,7), and $acc_0$ denotes the average accuracy of the conventional KNN without the $i^{th}$ feature set. After computing these weighted distances, the IKNN classifier sorts the distances in ascending order and selects the k training instances closest to the new instance. The classification decision in the IKNN classifier is based on a majority vote among the k-nearest neighbours. Thus, the IKNN classifier of the ISSBEC architecture enhances the conventional KNN approach by using the Weighted Minkowski distance to improve generalization, reduce overfitting, and achieve more accurate predictions in high-dimensional classification. Enhanced features from the MSE were combined to obtain the predicted outcome. By introducing a weighted Minkowski distance, the IKNN classifier enhanced the generalization ability of the ISSBEC architecture. This improvement ensures that the ISSBEC approach can better handle the complexities of high-dimensional data and achieve more accurate and reliable predictions. Algorithm

3 depicts the entire IKNN process. Finally, the predicted outcomes $P_1, P_2, ---, P_n$ from Block-1, Block-2, ____, Block-n are ensembled using majority voting and provide the final classification outcome, specified as $O_{ISSBEC}$. Although traditional ensemble fusion methods such as weighted voting, stacking, and probability-based fusion can improve predictive performance, they introduce notable drawbacks in high-dimensional environments. These methods are computationally expensive and prone to overfitting, and add complexity by increasing communication overhead in Spark-based systems. In contrast, majority voting is simple, robust, and computationally efficient, requiring no extra training. Its low communication cost and stability make it particularly advantageous for scalable ensemble learning in high-dimensional data settings. In this study, majority voting yielded a more predictable, interpretable, and manageable ensemble fusion strategy that conserves model diversity and is accurate for binary data. The framework was also tested on multi-class data.

## Results

The proposed high-dimensional data classification using the Spark framework was implemented using Python 3.7 on a 12th-generation Intel Core i5-1135G7 processor clocked at 2.4 GHz with 16GB RAM. To verify the effectiveness of our proposed model, we tested five datasets, i.e., Alizadeh-2000-v1 [71], Armstrong-2002-v1 [72], Chowdary-2006 [73], Golub-1999-v1 [74], and Gordon-2002 [75] respectively. The description of the datasets is provided in Table 3. All experiments were conducted using 80:20 train-test split across the benchmark data sets. Hyperparameters for the models were determined via 5-fold cross-validation on the training set to optimise model parameters. Averaged results from each run are then used as the basis for the metrics reported (accuracy, sensitivity, specificity, precision, f-measure, false negative rate, false positive rate, Matthews correlation coefficient, and negative predictive value), which were all calculated based on performance in the testing set.

A comprehensive analysis was conducted to evaluate the performance of the Spark framework-based ISSBEC approach by comparing it with the traditional methods. This analysis utilized an extensive set of metrics, including "Sensitivity, NPV, Specificity, F-measure, FNR, Precision, FPR, MCC, and accuracy," to provide a thorough analysis of the method's effectiveness in the ISSBEC. These evaluation metrics were mathematically presented through Eqs 23 to 31.

$$Sensitivity = \frac{TP}{TP + FN} \tag{23}$$

$$NegativePredictiveValue = \frac{TN}{TN + FN} \tag{24}$$

$$Specificity = \frac{TN}{TN + FP} \tag{25}$$

$$F-Measure = \frac{2 * Precision * Recall}{Precision + Recall} \tag{26}$$

**Table 3. Description of 5 benchmark datasets.**

| Dataset | Ref. | Samples | Attributes | Classes |
|---|---|---|---|---|
| Alizadeh2000-v1 | [71] | 42 | 1095 | 2 |
| Armstrong2002-v1 | [72] | 72 | 1081 | 2 |
| Chowdary2006 | [73] | 104 | 182 | 2 |
| Golub1999-v1 | [74] | 72 | 1868 | 2 |
| Gordon2002 | [75] | 181 | 1626 | 2 |

$$FalseNegativeRate = \frac{FN}{FN + TP} \tag{27}$$

$$Precision = \frac{TP}{TP + FP} \tag{28}$$

$$FalsePositiveRate = \frac{FP}{FP + TN} \tag{29}$$

$$MCC = \frac{(TN * TP) - (FN * FP)}{\sqrt{(TP + FP)(TP + FN)(TN + FP)(TN + FN)}} \tag{30}$$

$$Accuracy = \frac{(TP + TN)}{(TP + FP + FN + TN)} \tag{31}$$

Additionally, we included an ablation study, ROC curve analysis, and the Wilcoxon test. We comprehensively evaluated the efficacy of our proposed model using various metrics and conventional methods. The primary objective was to enhance the classification accuracy of the model.

Table 4 represents the comparative analysis of the ISSBEC strategy against several established methods, including EfficientNet, HDELC [31], CESE [43], KNN, LinkNet, SVM, RF and MvDSCN [55]. The analysis focuses on key performance metrics such as accuracy, sensitivity, specificity, precision, F-measure, Matthews correlation coefficient(MCC), NPV, false positive rate(FPR), and false negative rate(FNR) across benchmark datasets. With consistency across all datasets and metrics, the ISSBEC classifier is the most reliable. The ISSBEC provides accuracy values of over 0.95 for all the datasets, proving its capability for high-dimensional classification with more reliability. The sensitivity and specificity outcomes demonstrate the strength of the ISSBEC. The specificity records its score over 0.9, highlighting the ability to decrease false positives, while sensitivity scores beyond 0.9 indicate the minimal rate of false negatives. The significance of the F-measure shows that ISSBEC provides well-adjusted recall grouping and precision suitable for typical classification tasks. By achieving effective scores among all datasets, the MCC confirmed the strength of the ISSBEC approach. The ISSBEC produced significant NPV values, demonstrating its efficacy for identifying true negatives. On behalf of various error metrics, ISSBEC continuously achieves minimal FNR and FPR values. Its FNR values are ineffective with all methods, highlighting its ability to decrease missed positive conditions, while its FPR remains below 0.05, indicating a nominal rate of false positives. Table 4 shows that the ISSBEC demonstrates the enhanced performance compared to other classification methods across all datasets and metrics, establishing it as the most reliable approach. Its slight error rates, balanced specificity and sensitivity, and high accuracy indicate its efficiency over benchmark datasets, and it does not overfit the training data and performs consistently over folds and repetitions. Overfitting is controlled by the ensemble diversitys(Using FF-RSS and MSE) and feature selection(Using SVM-MRFE). This study proposes IDFC for data grouping, SVM-MRFE for feature selection, and ISSBEC for classification. An ablation study evaluated the impact of these approaches on our proposed model's performance, as shown in Table 5, and found that the proposed approach achieved better results in various metrics for all datasets. The performance of the proposed ISSBEC framework was used to analyse the sensitivity of its two most essential hyperparameters, the subspace dimensionality ratio $\gamma$ in the FF-RSS and the hybrid kernel weighting coefficient $\alpha$ in the IDFC, in terms of evaluating its robustness and reliability. The number of clusters (C) in the IDFC instance was fixed at C = 3, as this was the point at which all Spark nodes experienced balanced

**Table 4. Performance comparison of various models.**

| Dataset | Model | Accuracy | Sensitivity | Specificity | Precision | F-Measure | MCC | NPV | FPR | FNR |
|---|---|---|---|---|---|---|---|---|---|---|
| Alizadeh2000-v1 [71] | EfficientNet | 0.778 | 0.796 | 0.745 | 0.854 | 0.824 | 0.528 | 0.661 | 0.255 | 0.204 |
| | HDELC | 0.805 | 0.786 | 0.836 | 0.900 | 0.839 | 0.599 | 0.676 | 0.164 | 0.214 |
| | CESE | 0.785 | 0.786 | 0.782 | 0.871 | 0.827 | 0.550 | 0.662 | 0.218 | 0.214 |
| | KNN | 0.778 | 0.796 | 0.745 | 0.854 | 0.824 | 0.528 | 0.661 | 0.255 | 0.204 |
| | LinkNet | 0.785 | 0.767 | 0.818 | 0.888 | 0.823 | 0.562 | 0.652 | 0.182 | 0.233 |
| | SVM | 0.823 | 0.796 | 0.873 | 0.921 | 0.854 | 0.642 | 0.696 | 0.127 | 0.204 |
| | RF | 0.842 | 0.845 | 0.836 | 0.906 | 0.874 | 0.664 | 0.742 | 0.164 | 0.155 |
| | MvDSCN | 0.852 | 0.742 | 0.847 | 0.797 | 0.877 | 0.613 | 0.258 | 0.248 | 0.239 |
| | ISSBEC | 0.956 | 0.942 | 0.982 | 0.990 | 0.965 | 0.907 | 0.900 | 0.018 | 0.058 |
| Armstrong2002-v1 [72] | EfficientNet | 0.814 | 0.806 | 0.843 | 0.945 | 0.870 | 0.576 | 0.566 | 0.157 | 0.194 |
| | HDELC | 0.715 | 0.794 | 0.843 | 0.944 | 0.863 | 0.562 | 0.551 | 0.157 | 0.206 |
| | CESE | 0.792 | 0.782 | 0.824 | 0.937 | 0.853 | 0.533 | 0.532 | 0.176 | 0.218 |
| | KNN | 0.783 | 0.782 | 0.784 | 0.924 | 0.847 | 0.501 | 0.519 | 0.216 | 0.218 |
| | LinkNet | 0.769 | 0.794 | 0.686 | 0.894 | 0.841 | 0.435 | 0.500 | 0.314 | 0.206 |
| | SVM | 0.787 | 0.800 | 0.745 | 0.913 | 0.853 | 0.490 | 0.528 | 0.255 | 0.200 |
| | RF | 0.774 | 0.753 | 0.843 | 0.941 | 0.837 | 0.516 | 0.506 | 0.157 | 0.247 |
| | MvDSCN | 0.840 | 0.712 | 0.736 | 0.749 | 0.755 | 0.718 | 0.728 | 0.223 | 0.248 |
| | ISSBEC | 0.959 | 0.953 | 0.980 | 0.994 | 0.973 | 0.894 | 0.862 | 0.020 | 0.047 |
| Chowdary2006 [73] | EfficientNet | 0.867 | 0.857 | 0.878 | 0.889 | 0.873 | 0.734 | 0.844 | 0.122 | 0.143 |
| | HDELC | 0.715 | 0.726 | 0.703 | 0.735 | 0.731 | 0.429 | 0.693 | 0.297 | 0.274 |
| | CESE | 0.823 | 0.869 | 0.770 | 0.811 | 0.839 | 0.644 | 0.838 | 0.230 | 0.131 |
| | KNN | 0.759 | 0.798 | 0.716 | 0.761 | 0.779 | 0.516 | 0.757 | 0.284 | 0.202 |
| | LinkNet | 0.810 | 0.774 | 0.851 | 0.855 | 0.813 | 0.624 | 0.768 | 0.149 | 0.226 |
| | SVM | 0.778 | 0.786 | 0.770 | 0.795 | 0.790 | 0.556 | 0.760 | 0.230 | 0.214 |
| | RF | 0.810 | 0.774 | 0.851 | 0.855 | 0.813 | 0.624 | 0.768 | 0.149 | 0.226 |
| | MvDSCN | 0.860 | 0.765 | 0.737 | 0.751 | 0.757 | 0.502 | 0.752 | 0.263 | 0.235 |
| | ISSBEC | 0.956 | 0.952 | 0.959 | 0.904 | 0.928 | 0.911 | 0.947 | 0.041 | 0.048 |
| Golub1999 [74] | EfficientNet | 0.895 | 0.870 | 0.920 | 0.908 | 0.888 | 0.785 | 0.860 | 0.080 | 0.130 |
| | HDELC | 0.743 | 0.735 | 0.752 | 0.778 | 0.755 | 0.500 | 0.718 | 0.248 | 0.265 |
| | CESE | 0.829 | 0.813 | 0.845 | 0.865 | 0.839 | 0.650 | 0.800 | 0.155 | 0.187 |
| | KNN | 0.801 | 0.784 | 0.818 | 0.850 | 0.816 | 0.600 | 0.780 | 0.182 | 0.216 |
| | LinkNet | 0.850 | 0.835 | 0.865 | 0.878 | 0.856 | 0.700 | 0.820 | 0.135 | 0.165 |
| | SVM | 0.830 | 0.815 | 0.845 | 0.870 | 0.840 | 0.650 | 0.805 | 0.155 | 0.185 |
| | RF | 0.855 | 0.840 | 0.870 | 0.880 | 0.860 | 0.720 | 0.830 | 0.130 | 0.160 |
| | MvDSCN | 0.889 | 0.874 | 0.964 | 0.992 | 0.929 | 0.705 | 0.602 | 0.036 | 0.126 |
| | ISSBEC | 0.970 | 0.960 | 0.980 | 0.995 | 0.975 | 0.920 | 0.905 | 0.020 | 0.040 |
| Gordan2002 [75] | EfficientNet | 0.890 | 0.860 | 0.920 | 0.915 | 0.887 | 0.780 | 0.850 | 0.080 | 0.140 |
| | HDELC | 0.740 | 0.730 | 0.750 | 0.780 | 0.755 | 0.490 | 0.710 | 0.250 | 0.270 |
| | CESE | 0.825 | 0.810 | 0.840 | 0.860 | 0.835 | 0.640 | 0.790 | 0.160 | 0.190 |
| | KNN | 0.798 | 0.780 | 0.815 | 0.845 | 0.810 | 0.590 | 0.770 | 0.185 | 0.220 |
| | LinkNet | 0.848 | 0.830 | 0.865 | 0.875 | 0.850 | 0.690 | 0.810 | 0.135 | 0.170 |
| | SVM | 0.828 | 0.812 | 0.845 | 0.868 | 0.838 | 0.648 | 0.803 | 0.155 | 0.188 |
| | RF | 0.850 | 0.835 | 0.865 | 0.878 | 0.855 | 0.710 | 0.820 | 0.135 | 0.165 |
| | MvDSCN | 0.825 | 0.731 | 0.718 | 0.754 | 0.742 | 0.447 | 0.692 | 0.282 | 0.269 |
| | ISSBEC | 0.968 | 0.958 | 0.978 | 0.992 | 0.970 | 0.915 | 0.900 | 0.022 | 0.042 |

**Table 5. Ablation analysis on ISSBEC, model with conventional DFC, and model with conventional RFE.**

| Metrics | Alizadeh2000-v1 | | | Armstrong2002-v1 | | | Chowdary2006 | | | Golub1999-v1 | | | Gordon2002 | | |
|---|---|---|---|---|---|---|---|---|---|---|---|---|---|---|---|
| | DFC | RFE | ISSBEC | DFC | RFE | ISSBEC | DFC | RFE | ISSBEC | DFC | RFE | ISSBEC | DFC | RFE | ISSBEC |
| Accuracy | 0.787 | 0.790 | 0.962 | 0.804 | 0.787 | 0.955 | 0.787 | 0.784 | 0.956 | 0.781 | 0.768 | 0.959 | 0.790 | 0.792 | 0.965 |
| Sensitivity | 0.787 | 0.803 | 0.941 | 0.797 | 0.782 | 0.959 | 0.819 | 0.782 | 0.952 | 0.780 | 0.773 | 0.928 | 0.793 | 0.789 | 0.928 |
| Specificity | 0.789 | 0.766 | 0.934 | 0.825 | 0.802 | 0.941 | 0.757 | 0.786 | 0.959 | 0.786 | 0.743 | 0.886 | 0.788 | 0.796 | 0.906 |
| Precision | 0.880 | 0.871 | 0.940 | 0.938 | 0.929 | 0.902 | 0.763 | 0.782 | 0.904 | 0.957 | 0.945 | 0.940 | 0.815 | 0.815 | 0.927 |
| FPR | 0.211 | 0.234 | 0.036 | 0.175 | 0.198 | 0.059 | 0.243 | 0.214 | 0.041 | 0.214 | 0.257 | 0.114 | 0.212 | 0.204 | 0.094 |
| FNR | 0.213 | 0.197 | 0.039 | 0.203 | 0.218 | 0.041 | 0.181 | 0.218 | 0.048 | 0.220 | 0.227 | 0.022 | 0.207 | 0.211 | 0.042 |
| F-Measure | 0.831 | 0.836 | 0.931 | 0.862 | 0.849 | 0.920 | 0.790 | 0.782 | 0.928 | 0.859 | 0.850 | 0.934 | 0.803 | 0.802 | 0.942 |
| MCC | 0.554 | 0.551 | 0.917 | 0.552 | 0.516 | 0.877 | 0.577 | 0.567 | 0.911 | 0.430 | 0.398 | 0.873 | 0.579 | 0.583 | 0.868 |
| NPV | 0.653 | 0.663 | 0.930 | 0.552 | 0.528 | 0.873 | 0.815 | 0.786 | 0.947 | 0.370 | 0.363 | 0.912 | 0.764 | 0.767 | 0.946 |

data distribution and stable partitioning. All subsequent studies were conducted with C = 3, as selecting C = 2–4 during the evaluation phase yielded negligible performance differences across the three values. In contrast, higher cluster counts were associated with longer computation times and not much variance in performance accuracy. The sensitivity analysis varied $\gamma \in [0.2, 0.8]$ and $\alpha \in [0.0, 1.0]$, which can be reprasented in Fig 2 and from the figure it is observed that the accuracy remains within ±2% for every change in a parameter. This demonstrates that even when parameters and data sets are changed, the framework's performance remains steady and consistent. This demonstrates ISSBEC's scalability and resilience.

The ROC analysis for the ISSBEC model was compared with EfficientNet, HDELC [31], CESE [43], KNN, LinkNet, SVM, and RF for high-dimensional data classification in the Spark framework. This comparison across benchmark datasets is illustrated in Figs 3, 4, 5, 6, and 7, which shows the performance of each model. From Figs 3, 4, 5, 6, and 7, it is evident that the ISSBEC strategy demonstrated superior AUC performance in contrast to traditional approaches. This consistent superiority in AUC across benchmark datasets underscores the effectiveness of the ISSBEC strategy for high-dimensional data classification, demonstrating its ability to achieve better classification performance and greater overall effectiveness compared to conventional methods. This achievement is attributed to the strategic integration of IDFC for effective data partitioning, the application of SVM-MRFE for effective FS, and the utilization of an improved ensemble

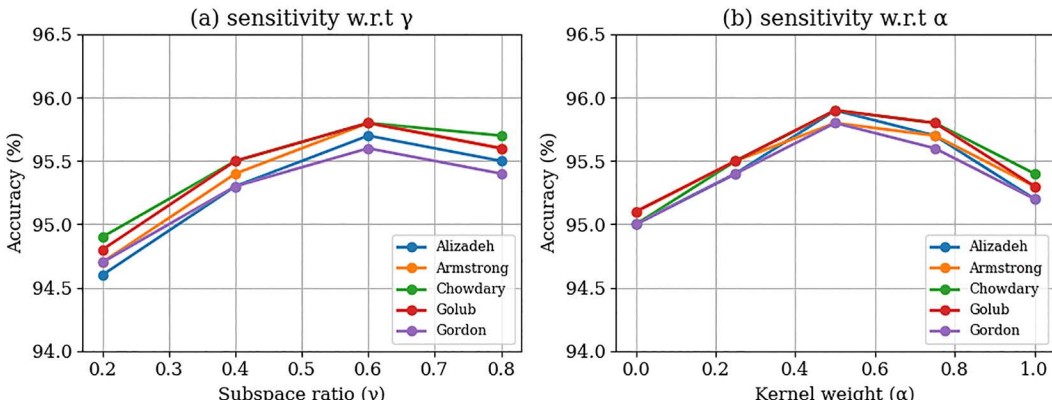

**Fig 2. Sensitivity Analysis.**

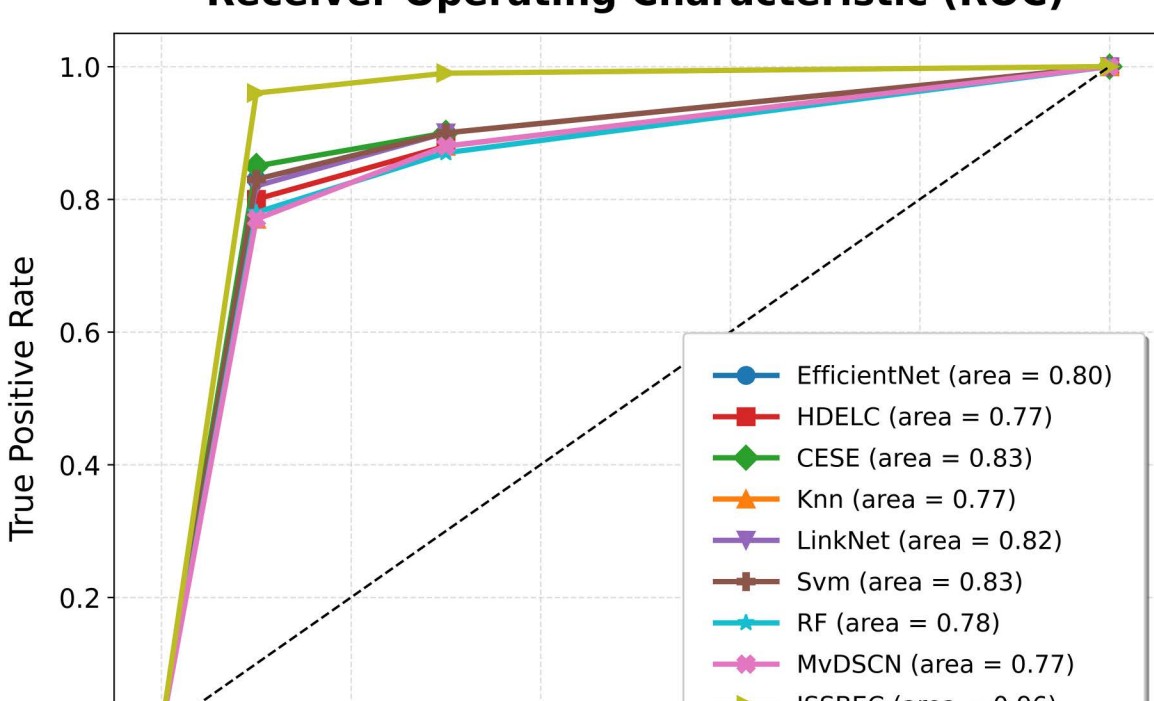

**Fig 3. ROC Analysis for ISSBEC and Conventional Methods while using datasets a) Alizadeh2000-v1.**

model combining SVM, RF, and IKNN, which together enhance the model's classification capabilities and yield enhanced ROC performance.

The Friedman test is a non-parametric statistical test used to compare three or more related groups, tested on the same datasets. It ranks the performance of each method within each dataset and calculates whether the differences in ranks are statistically significant. A p-value below 0.05 indicates a significant difference between methods. Table 6 presents the results of the Friedman test comparing the performance of ISSBEC with various classifiers across five datasets. The p-values indicate that, in most cases, ISSBEC performs significantly differently from the other models. In comparison with EfficientNet shows p-values ranging from 0.044 to 0.077, HDELC ranges from 0.020 to 0.073. Other models, such as CESE, KNN, LinkNet, SVM, RF, and MvDSCN, also show varying levels of significance across datasets, with several p-values below the 0.05 threshold, suggesting statistically significant differences. Overall, the Friedman test confirms that ISSBEC consistently exhibits distinct and superior performance compared to the majority of existing methods across all datasets.

The Wilcoxon test results in Table 7 show how the performance of ISSBEC compares statistically with other classifiers across five datasets. Using a significance threshold of p-value less than 0.1, several models display statistically significant differences when compared with ISSBEC. Moreover, the ISSBEC shows significant differences against CESE on Alizadeh2000-v1 (p = 0.051) and SVM on Alizadeh2000-v1 (p = 0.054). In Armstrong2002-v1, HDELC (p = 0.070), KNN (p = 0.075), SVM (p = 0.092), RF (p = 0.107, not significant). Across other datasets, multiple

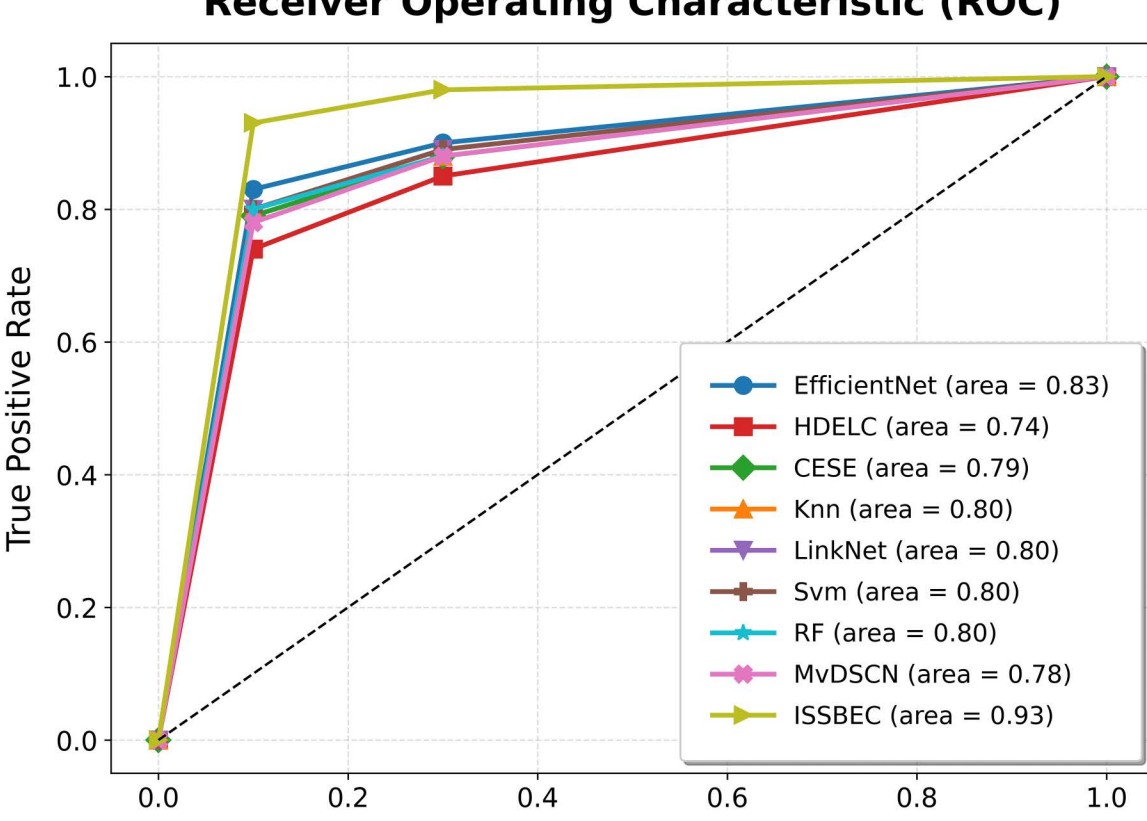

## Receiver Operating Characteristic (ROC)

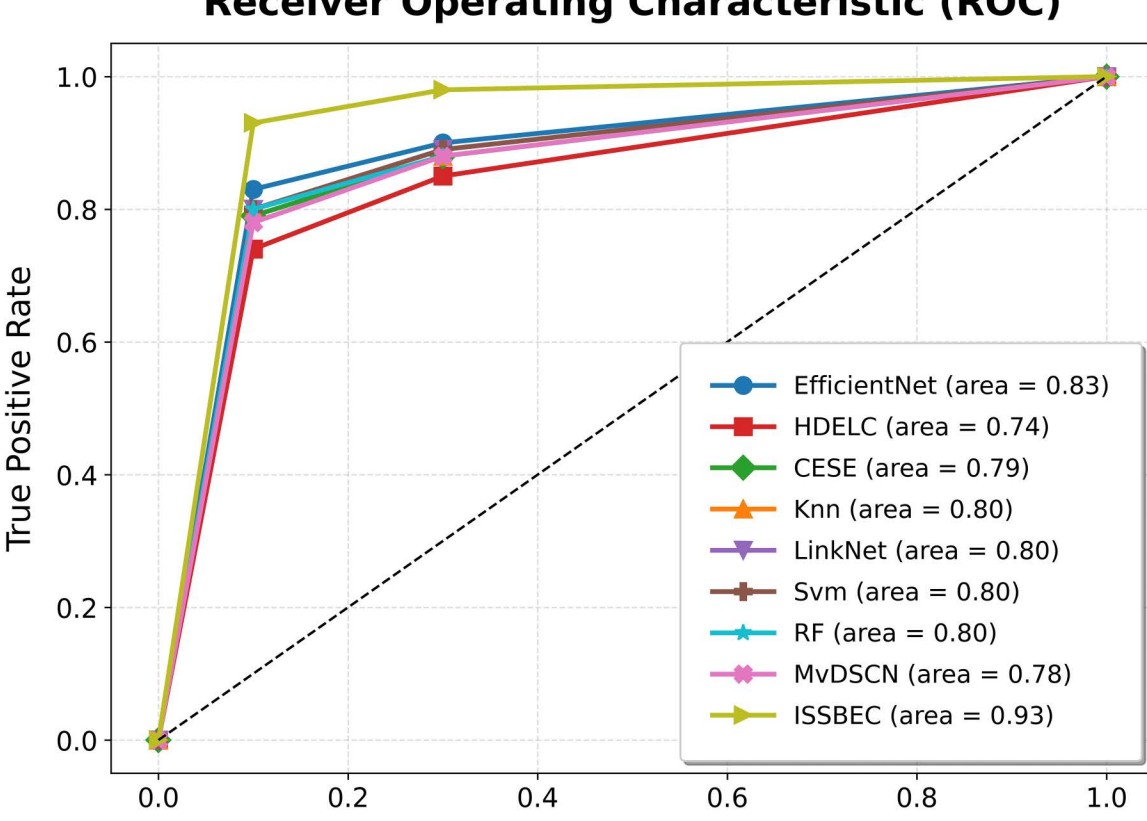

**Fig 4. ROC Analysis for ISSBEC and Conventional Methods while using datasets b) Armstrong2002-v1.**

models, including LinkNet, and MvDSCN, record p-values close to below the 0.1 threshold, indicating consistent statistical separation between ISSBEC and competing methods. Overall, the Wilcoxon analysis confirms that ISS-BEC's performance differs significantly from most existing classifiers on various datasets, supporting its robustness and superiority.

All the benchmark datasets have binary class labels, and our model was also tested with two multiclass label datasets with 3 and 10 as class labels, experimented in [43]. These experimental results are shown in Table 8, from which it is observed that our ISSBEC approach also shows improvement on multi-class label datasets. In future we extend our model for handling multi-class label as well as extreme high-dimensional data.

## Complexity analysis

This section presents a detailed, explicit discussion of the theoretical analysis of the time complexity of the proposed spark-based and non-spark-based ISSBECs for the high-dimensional data classification problem. The model encompasses data preprocessing, IDFC clustering, SVM-MRFE feature selection, feature fusion, subspace transformation (FF-RSS + MSE), base-learner classification (SVM, RF, IKNN), and ensemble voting.

The non-spark (single machine) version performs all computations sequentially. Essential procedures like IDFC and SVM-MRFE have quadratic time complexity that depends on the size of the data and the feature dimension.

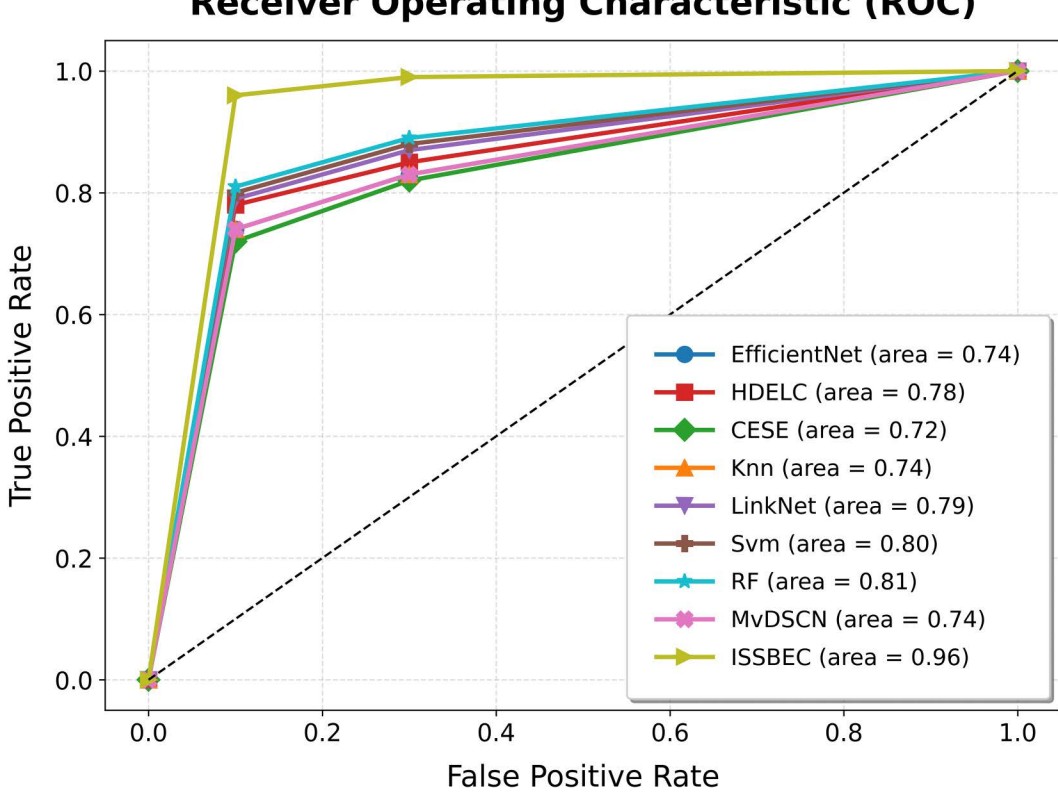

**Fig 5. ROC Analysis for ISSBEC and Conventional Methods while using datasets c) Chowdary2006.**

This leads to an infeasibly long execution time even for small-sized high-dimensional datasets. On the other hand, the Spark implementation uses $P$ partitions, leading to sublinear-time improvements in the processing modules. Significant performance improvements are achieved without any degradation in accuracy by parallelising feature selection, clustering, and subspace generation. The stage-wise complexity analysis of both phases is mentioned in Table 9. Where, $n$ denotes the total number of data samples, $d$ denotes the number of features, $k$ denotes the number of clusters in IDFC, $m$ denotes the number of features per subspace, and $T$ denotes the number of decision trees in the random forest, $l$ denotes the number of subspace rotations in MSE and $C$ denotes the classifier.

## Discussion

The proposed ISSBEC framework integrates IDFC partitioning and SVM-MRFE feature selection to address the challenges of high-dimensional data classification. The ablation analysis clearly shows that substituting either component with conventional methods leads to substantial performance degradation, confirming that both IDFC and SVM-MRFE contribute uniquely to the overall effectiveness of the system. The experimental results across all five benchmark datasets (Alizadeh2000-v1, Armstrong2002-v1, Chowdary2006, Golub1999, and Gordan2002) consistently demonstrate the superior performance of the proposed ISSBEC classifier compared to existing models such as KNN, SVM, RF, Efficient-Net, HDELC, and CESE, showing varying degrees of effectiveness but struggling to maintain robustness and stability across high-dimensional datasets. Specifically, the ISSBEC records the highest accuracy, ranging from 0.956 to 0.970,

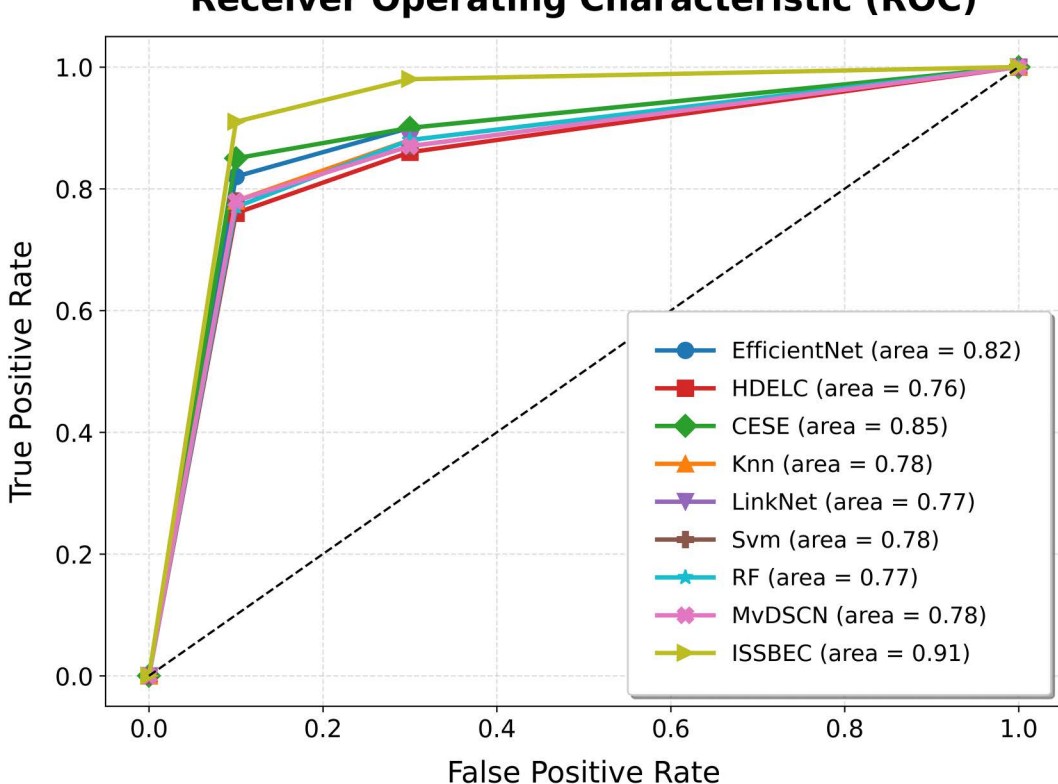

**Fig 6. ROC Analysis for ISSBEC and Conventional Methods while using datasets d) Golub1999-v1.**

significantly outperforming the compared models, which generally achieve accuracies between 0.74 and 0.89. Moreover, ISSBEC's precision (0.904–0.994) and F-measure (0.928–0.975) indicate that the proposed framework achieves both high true-positive rates and excellent classification balance. Despite these strengths, its performance on multimodal data remains unexplored. Nevertheless, these findings reinforce the importance of combining advanced clustering with robust feature selection in ensemble-based learning.

The model proposed in this paper conforms its suitability for binary classification. However, we tried to verify this approach with multi class datasets and found a minor difference while comparing some baseline such as CESE in Table 8. Hence, we would like to extend the framework suitably for multi class and multi-modal high-dimensional data classification with improved accuracy and less complexity in future.

## Conclusion

A novel high-dimensional classification approach is proposed in this study by combining the Spark framework with the ISSBEC architecture. The approach is divided into three main stages: preprocessing, processing through the Spark framework, and classification. During preprocessing, the high-dimensional data were min-max normalized to standardize them. The Spark framework was used to handle high-dimensional data, with the master node performing IDFC-based data partitioning and the slave node performing FS with SVM-MRFE, followed by feature fusion. The classification stage was employed in the ISSBEC approach, which requires several blocks, each block featuring FF-RSS, MSE, and a base classifier. We used RF in Block-1, SVM in Block-2, and IKNN in Block-3 as the base

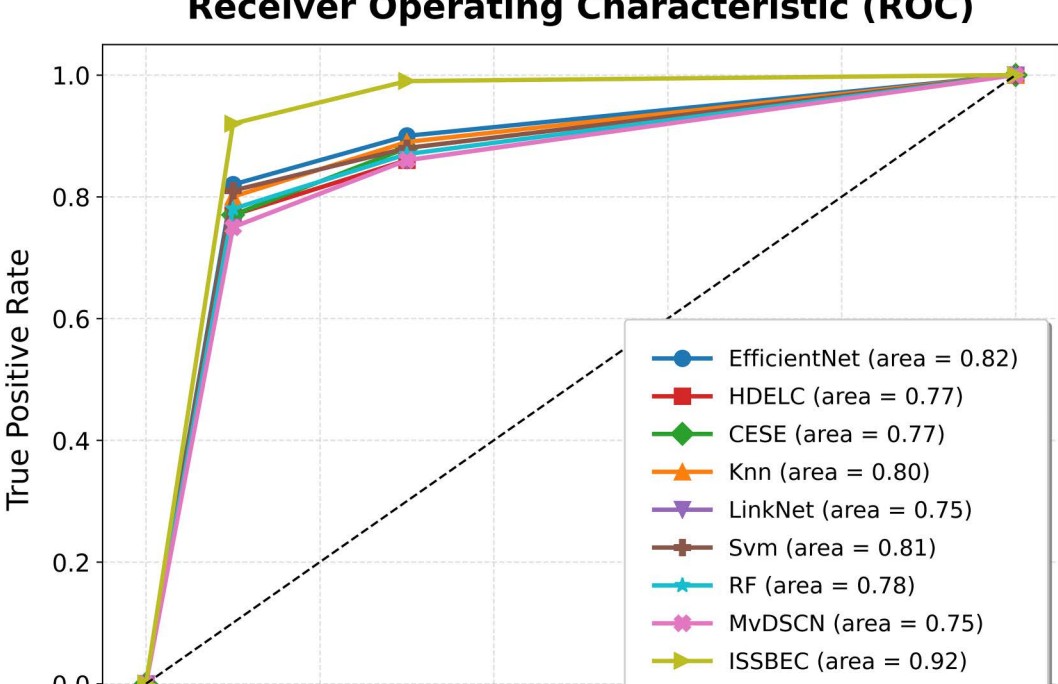

**Fig 7. ROC Analysis for ISSBEC and Conventional Methods while using datasets e) Gordon2002.**

**Table 6. Friedman test analysis.**

| ISSBEC vs. | P-value | | | | |
|---|---|---|---|---|---|
| | Alizadeh2000-v1 | Armstrong2002-v1 | Chowdary2006 | Golub1999-v1 | Gordon2002 |
| EfficientNet | 0.074 | 0.077 | 0.045 | 0.073 | 0.044 |
| HDELC | 0.073 | 0.030 | 0.020 | 0.073 | 0.060 |
| CESE | 0.033 | 0.033 | 0.077 | 0.029 | 0.019 |
| KNN | 0.080 | 0.048 | 0.044 | 0.086 | 0.068 |
| LinkNet | 0.042 | 0.057 | 0.043 | 0.086 | 0.026 |
| SVM | 0.089 | 0.056 | 0.035 | 0.033 | 0.016 |
| RF | 0.054 | 0.046 | 0.072 | 0.065 | 0.078 |
| MvDSCN | 0.033 | 0.052 | 0.083 | 0.053 | 0.032 |

classifiers. The proposed approach was evaluated using several performance metrics and compared with existing methods, demonstrating significant improvements in the classification performance and efficacy of high-dimensional classification. In this study, we designed the proposed model for small-scale, high-dimensional data with continuous features and binary class labels. Further, the proposed model may be extended suitably to classify multi-class and multi-model high dimensional datasets.

**Table 7. Wilcoxon test analysis.**

| ISSBEC Vs | Alizadeh2000-v1 | Armstrong2002-v1 | Chowdary2006 | Golub1999-v1 | Gordon2002 |
|---|---|---|---|---|---|
| EfficientNet | 0.12 | 0.105 | 0.116 | 0.117 | 0.106 |
| HDELC | 0.10 | 0.07 | 0.08 | 0.093 | 0.124 |
| CESE | 0.051 | 0.101 | 0.105 | 0.115 | 0.096 |
| Knn | 0.119 | 0.075 | 0.089 | 0.112 | 0.106 |
| LinkNet | 0.092 | 0.091 | 0.072 | 0.123 | 0.097 |
| Svm | 0.054 | 0.092 | 0.13 | 0.118 | 0.095 |
| RF | 0.118 | 0.107 | 0.096 | 0.092 | 0.088 |
| MvDSCN | 0.111 | 0.11 | 0.112 | 0.08 | 0.11 |

**Table 8. Performance comparison in terms of accuracy while using multi-class label datasets.**

| Dataset | Classes | CESE [43] | ISSBEC (Proposed Model) |
|---|---|---|---|
| Su [76] | 10 | 0.9024 | 0.8942 |
| Bredel [77] | 3 | 0.848 | 0.8880 |

**Table 9. Time complexity comparison of non-Spark and Spark-based ISSBEC framework.**

| Stage | Non-Spark Based ISSBEC | Spark Based ISSBEC |
|---|---|---|
| Preprocessing | $\mathcal{O}(nd)$ | $\mathcal{O}(nd)$ |
| Data Partitioning | $\mathcal{O}(n^2d + knd)$ | $\mathcal{O}\left(\frac{n^2d+knd}{P}\right)$ |
| Feature Selection | $\mathcal{O}(nd^2)$ | $\mathcal{O}\left(\frac{nd^2}{P}\right)$ |
| Feature Fusion | $\mathcal{O}(nd)$ | $\mathcal{O}\left(\frac{nd}{P}\right)$ |
| FF-RSS | $\mathcal{O}(m^2)$ | $\mathcal{O}\left(\frac{m^2}{P}\right)$ |
| MSE | $\mathcal{O}(ml)$ | $\mathcal{O}\left(\frac{ml}{P}\right)$ |
| Base Classifiers | SVM $\rightarrow \mathcal{O}(n^2)$ | SVM $\rightarrow \mathcal{O}(n^2)$ |
|  | RF $\rightarrow \mathcal{O}(Tn\log n)$ | RF $\rightarrow \mathcal{O}(Tn\log n)$ |
|  | IKNN $\rightarrow \mathcal{O}(nd)$ | IKNN $\rightarrow \mathcal{O}(nd)$ |
| Ensemble | $\mathcal{O}(n)$ | $\mathcal{O}(n)$ |
| Total | $\mathcal{O}(nd + n^2d + knd$ $+nd^2 + m^2 + ml + C + n)$ | $\mathcal{O}\left(nd + \frac{n^2d+knd+nd^2}{P}\right.$ $\left. +m^2 + ml + C + n\right)$ |

# Author contributions

**Conceptualization:** Venkaiah Chowdary Bhimineni, Rajiv Senapati.

**Data curation:** Venkaiah Chowdary Bhimineni.

**Formal analysis:** Venkaiah Chowdary Bhimineni, Rajiv Senapati.

**Investigation:** Venkaiah Chowdary Bhimineni, Rajiv Senapati.

**Methodology:** Venkaiah Chowdary Bhimineni, Rajiv Senapati.

**Project administration:** Venkaiah Chowdary Bhimineni, Rajiv Senapati.

**Resources:** Venkaiah Chowdary Bhimineni.

**Software:** Venkaiah Chowdary Bhimineni.

**Supervision:** Rajiv Senapati.

**Validation:** Venkaiah Chowdary Bhimineni, Rajiv Senapati.

**Visualization:** Venkaiah Chowdary Bhimineni.

**Writing – original draft:** Venkaiah Chowdary Bhimineni.

**Writing – review & editing:** Venkaiah Chowdary Bhimineni, Rajiv Senapati.

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
