## [Decision Letter · Decision Letter 0]

12 Nov 2025

Dear Dr. Senapati,

Thank you for submitting your manuscript to PLOS ONE. After careful consideration, we feel that it has merit but does not fully meet PLOS ONE’s publication criteria as it currently stands. Therefore, we invite you to submit a revised version of the manuscript that addresses the points raised during the review process.

We look forward to receiving your revised manuscript.

Kind regards,

Razieh Sheikhpour

Academic Editor

PLOS ONE

Reviewers' comments:

Reviewer's Responses to Questions

**Comments to the Author**

1. Is the manuscript technically sound, and do the data support the conclusions?

Reviewer #1: Yes

Reviewer #2: Yes

2. Has the statistical analysis been performed appropriately and rigorously?

Reviewer #1: Yes

Reviewer #2: Yes

3. Have the authors made all data underlying the findings in their manuscript fully available?

Reviewer #1: Yes

Reviewer #2: Yes

4. Is the manuscript presented in an intelligible fashion and written in standard English?

Reviewer #1: Yes

Reviewer #2: Yes

Reviewer #1: This manuscript presents ISSBEC, a Spark-based ensemble classifier for high-dimensional data classification, incorporating min-max normalization, improved deep fuzzy clustering (IDFC) for partitioning, SVM-modified recursive feature elimination (SVM-MRFE) for selection, and an improved subspace-based ensemble with feature fusion random subspace (FF-RSS), mixed space enhancement (MSE), and diverse base classifiers to tackle the curse of dimensionality. Evaluated on datasets via metrics like accuracy and robustness, it claims superiority over state-of-the-art methods. While offering a scalable solution for overfitting and sparsity in big data environments, the work requires major revisions to address unclear novelty amid recent Spark-ensemble hybrids, incomplete methodological proofs, narrow experimental scope, and significant presentation flaws for PLOS ONE publication.

The manuscript is riddled with grammatical errors, repetitive phrasing (e.g., "high-dimensional data" overused without variation), and unclear sections with abrupt transitions, severely affecting readability; a thorough professional edit is mandatory to enhance clarity and academic tone.

The related work section lacks depth, ignoring 2024-2025 advancements like Spark-based ensemble for high-dimensional anomaly detection and adaptive subspace ensembles for imbalanced data; include a comparison table to highlight unique FF-RSS and MSE contributions.

The IDFC partitioning and SVM-MRFE selection are innovative but lack theoretical justification; provide proofs on convergence and optimality under high sparsity, comparing to standard fuzzy C-means or RFE variants.

FF-RSS and MSE mechanisms are vaguely described, risking instability in distributed environments; elaborate with detailed equations, pseudocode, and analyses of fusion impacts on ensemble diversity.

Parameter choices for clustering depth and subspace sizes are not analyzed; conduct sensitivity studies to show effects on performance metrics across varying dimensions.

Illustrative examples are absent; add toy datasets demonstrating partitioning, feature fusion, and ensemble steps to clarify the workflow.

Experimental datasets are standard but lack diversity (e.g., no real-time streaming or multi-modal data); test on broader benchmarks from recent Spark ML surveys to validate scalability.

Baseline comparisons are limited; incorporate recent methods like deep ensemble subspace clustering, with statistical tests (e.g., Friedman) over multiple runs for rigor.

The Literature citation is not adequate, and the related work to machine learning should be discussed

1.Artificial intelligence in multimodal learning analytics: a systematic literature review

2.NEDL-GCP: A nested ensemble deep learning model for Gynaecological cancer risk prediction

Reviewer #2: The manuscript presents an ensemble learning framework based on random subspaces integrated with feature selection and feature fusion using the Spark platform. The topic is relevant and potentially valuable for high-dimensional data analysis. However, the manuscript, in its current form, has several critical weaknesses in methodological clarity, experimental completeness, and presentation structure. These issues prevent the reader from fully understanding and evaluating the novelty and validity of the proposed approach.

Strengths:

- The paper addresses a significant problem: classification of high-dimensional data and scalability using distributed computing.

- The proposed integration of Spark with ensemble learning and subspace-based methods could be promising if properly justified and evaluated.

- The manuscript is written in generally clear language and follows a standard scientific format.

Major Weaknesses:

- Ambiguity in Feature Selection (SVM-MRFE):

The description of the hybrid feature selection method is unclear. It is not explained how the Modified Fisher Score and SVM-RFE are combined or how the ranking is performed. The choice of a backward elimination strategy (wrapper-based) introduces risks of local optima and poor search diversity.

Additionally, the use of a linear SVM as the wrapper classifier is questionable for non-linear data. The authors should justify this design choice and discuss alternatives such as metaheuristic feature selection algorithms or kernel-based SVMs.

- Unclear Purpose of Feature Fusion:

The rationale for performing feature fusion after feature selection is not convincing. Once the optimal subset of features has been selected, further dimensionality reduction using PCA may result in information loss. The authors should explain the purpose of this step, quantify the retained variance, and demonstrate its impact on performance.

- Weak Methodology Organization:

The methodology section is fragmented and repetitive. A clearer flow is needed—each stage should be fully described once, in logical sequence, rather than being introduced briefly and elaborated later.

- Simplistic Ensemble Voting Strategy:

The use of majority voting for combining classifiers is too simplistic and lacks justification. More robust approaches (e.g., weighted voting, stacking, or probability-based fusion) could lead to better performance, especially for multi-class scenarios. The authors should explain why this basic method was chosen and discuss its limitations.

- Incomplete Evaluation on Multi-Class Data:

The experiments are primarily limited to binary datasets. The multi-class evaluation is minimal, with only two datasets and incomplete comparisons. To substantiate the generalization capability of the proposed model, more multi-class datasets and comparative studies are required.

- Lack of Evaluation Procedure Details:

The paper does not specify how results were obtained (training/test split, cross-validation strategy, or prevention of overfitting). It is unclear whether reported metrics correspond to test or training results. A detailed evaluation methodology must be provided for reproducibility.

- Model Complexity vs. Performance Trade-off:

The proposed system is highly complex, combining several components (IDFC, SVM-MRFE, Feature Fusion, ISSBEC). However, Table 8 indicates only modest performance improvements over simpler methods. The authors should justify whether this level of complexity is necessary.

- Absence of a Discussion Section:

The manuscript lacks a Discussion section, which is essential to interpret results, highlight contributions, discuss limitations, and relate findings to existing literature.

Minor Comments:

- Improve figure references and algorithm readability (some symbols are not defined in context).

- Clarify parameter settings and computational complexity in the experimental section.

- Ensure consistency in notation (e.g., SVM-MRFE and MRFE are used interchangeably).

- Revise grammatical issues in long sentences for better readability.

**Do you want your identity to be public for this peer review?** For information about this choice, including consent withdrawal, please see our Privacy Policy

Reviewer #1: No

Reviewer #2: **Yes:** Amin Hashemi

---

## [Author Response · Author response to Decision Letter 1]

11 Dec 2025

Reviewer #1:

Query 1. The manuscript is riddled with grammatical errors, repetitive phrasing (e.g., "high-dimensional data" overused without variation), and unclear sections with abrupt transitions, severely affecting readability; a thorough professional edit is mandatory to enhance clarity and academic tone.

Reply: As per the suggestion, the manuscript is modified suitably.

Query 2. The related work section lacks depth, ignoring 2024-2025 advancements like Spark-based ensembles for high-dimensional anomaly detection and adaptive subspace ensembles for imbalanced data; include a comparison table to highlight unique FF-RSS and MSE contributions.

Reply: As per the suggestion, the literature review section is improved suitably. Further, the comparison of proposed approach with the recent state-of-the-art is presented in Table 1, available on Page 5.

Query 3. The IDFC partitioning and SVM-MRFE selection are innovative but lack theoretical justification; provide proofs on convergence and optimality under high sparsity, comparing to standard fuzzy C-means or RFE variants.

Reply: As per the reviewer suggestion the rationale behind the choice of using IDFC has been mentioned in the first paragraph of the section “Data partitioning in Master Node”, which can be found in Page 7 and the rationale behind the choice of using SVM-MRFE has been mentioned in the subsection “Feature Selection by Modified Recursive Feature Elimination”, which can be found in Page 9-10. The convergence justification is provided at the end of the section “Data partitioning in Master Node”, can be found on Page 9.

Query 4. FF-RSS and MSE mechanisms are vaguely described, risking instability in distributed environments; elaborate with detailed equations, pseudocode, and analyses of fusion impacts on ensemble diversity.

Reply: As per the reviewers' suggestion, we updated the FF-RSS and MSE subsections available in Page 12 and 13.

Query 5. Parameter choices for clustering depth and subspace sizes are not analysed; conduct sensitivity studies to show effects on performance metrics across varying dimensions.

Illustrative examples are absent; add toy datasets demonstrating partitioning, feature fusion, and ensemble steps to clarify the workflow.

Reply: The proper justification of parameter choice for clustering depth is now suitably presented in the revised version of this manuscript, which can be found on page 16-18. Further, as per the advice we have conducted sensitivity analysis, which can be found from Figure 2, available on page 18 .

Query 6. Experimental datasets are standard but lack diversity (e.g., no real-time streaming or multi-modal data); test on broader benchmarks from recent Spark ML surveys to validate scalability.

Reply: The model proposed in this paper is also tested with multi-class dataset. The experimental results with baseline comparison are presented in Table 8 in the revised version of this manuscript.

Query 7. Baseline comparisons are limited; incorporate recent methods like deep ensemble subspace clustering, with statistical tests (e.g., Friedman) over multiple runs for rigour.

Reply: As pe the reviewer suggestion we have included recent baseline models while comparing our proposed model with them. It can be found in Table 4, available on page 17.

Query 8. The Literature citation is not adequate, and the related work to machine learning should be discussed “1.Artificial intelligence in multimodal learning analytics: a systematic literature review, 2.NEDL-GCP: A nested ensemble deep learning model for Gynaecological cancer risk prediction”.

Reply: As per the recommendation, the literature section is updated suitably.

Reviewer #2:

Query 1. The description of the hybrid feature selection method is unclear. It is not explained how the Modified Fisher Score and SVM-RFE are combined or how the ranking is performed. The choice of a backward elimination strategy (wrapper-based) introduces risks of local optima and poor search diversity.

Reply: The proposed work uses SVM-MRFE approach for selecting the optimal feature. The SVM-MRFE approach is the improved version of the conventional SVM-RFE approach. The proposed SVM-MRFE uses the modified Fisher score, as mentioned in Eq. (14) and ranks each feature based on this Fisher score; here, the feature having the smallest ranking value gets eliminated. The detailed description is added in the section “Feature Selection by Modified recursive Feature Elimination”, which can be found on Page 9-10 of the revised manuscript.

Query 2. Additionally, the use of a linear SVM as the wrapper classifier is questionable for non-linear data. The authors should justify this design choice and discuss alternatives such as metaheuristic feature selection algorithms or kernel-based SVMs.

Reply: As per the suggestion, the rationale behind the choice of using the SVM classifier has been theoretically mentioned in the section “Importance of Base Classifiers” available in page 13. Further, the use of every components are validated through ablation analysis available in Table 5 on page 18, where the model omits the SVM-MRFE model and uses the conventional RFE, shows lower performance in comparison to the proposed ISSBEC approach.

Query 3. The rationale for performing feature fusion after feature selection is not convincing. Once the optimal subset of features has been selected, further dimensionality reduction using PCA may result in information loss. The authors should explain the purpose of this step, quantify the retained variance, and demonstrate its impact on performance.

Reply: The feature selection reduces dimensionality by retaining only the most informative features; redundancy may still exist among the selected features. Feature fusion combines these selected features into a compact form, reducing redundancy and improving the stability and robustness of the classifier. This step can enhance generalisation and lead to improved classification performance. The importance of employing the feature fusion approach after the feature selection process is now theoretically discussed in Page 11. Further, we have conducted ablation analysis to conform the same.

Query 4. The methodology section is fragmented and repetitive. A clearer flow is needed—each stage should be fully described once, in logical sequence, rather than being introduced briefly and elaborated later.

Reply: The revised version of this manuscript is modified suitably.

Query 5. The use of majority voting for combining classifiers is too simplistic and lacks justification. More robust approaches (e.g., weighted voting, stacking, or probability-based fusion) could lead to better performance, especially for multi-class scenarios. The authors should explain why this basic method was chosen and discuss its limitations.

Reply: As per the reviewer’s suggestion, the rationale behind the choice of using the majority voting approach has been mentioned at the end of the section “Importance of Base Classifiers”, which can be found on Page 13-15 of the revised manuscript. Unlike the existing ensemble fusion methods such as weighted voting, stacking, or probability-based fusion, the majority voting strategy is simple, robust and does not require any extra training and is highly advantageous in high-dimensional data settings.

Query 6. The experiments are primarily limited to binary datasets. The multi-class evaluation is minimal, with only two datasets and incomplete comparisons. To substantiate the generalization capability of the proposed model, more multi-class datasets and comparative studies are required.

Reply: The model proposed in this paper conforms its suitability for binary classification. However, we tried to verify this approach with multi class datasets and found a minor difference while comparing some baseline such as CESE in Table 8. Hence, we would like to extend the framework suitably for multi class classification with improved accuracy and less complexity in future.

Query 7. The paper does not specify how results were obtained (training/test split, cross-validation strategy, or prevention of overfitting). It is unclear whether reported metrics correspond to test or training results. A detailed evaluation methodology must be provided for reproducibility.

Reply: As per the reviewer's suggestion, the results section has been updated with corresponding information, and it can be found on Page 15 of the revised manuscript.

Query 8. The proposed system is highly complex, combining several components (IDFC, SVM-MRFE, Feature Fusion, ISSBEC). However, Table 8 indicates only modest performance improvements over simpler methods. The authors should justify whether this level of complexity is necessary.

Reply: The Table 8 is now Table 9 in the revised version of this manuscript. Through this table we have clearly demonstrated the complexity comparison between non spark and spark based framework. The details description is now presented in Complexity analysis section available in Page 20 and 21.

Query 9. The manuscript lacks a Discussion section, which is essential to interpret results, highlight contributions, discuss limitations, and relate findings to existing literature.

Reply: As per the reviewer’s suggestion, the Discussion section has been included in the revised manuscript, which can be found on Page 21.

Query 10. Improve figure references and algorithm readability (some symbols are not defined in context).

Reply: As per the reviewer’s suggestion, we have improved the references of figures, algorithms and all variables defined in the revised version of this manuscript.

Query 11. Clarify parameter settings and computational complexity in the experimental section.

Reply: The result section is suitably modified in the revised version of this manuscript. Further, the modified computational complexity analysis can be found on Page 20-21.

Query 12. Ensure consistency in notation (e.g., SVM-MRFE and MRFE are used interchangeably).

Reply: As per the reviewer's suggestion, the terminologies have been used consistently in the revised manuscript.

Query 13. Revise grammatical issues in long sentences for better readability.

Reply: As per the direction of reviewers’ the manuscript is revised suitably.

---

## [Decision Letter · Decision Letter 1]

22 Jan 2026

Random Subspace-Based Ensemble Classifier for High-Dimensional Data Using SPARK

PONE-D-25-47457R1

Dear Dr. Senapati,

We’re pleased to inform you that your manuscript has been judged scientifically suitable for publication and will be formally accepted for publication once it meets all outstanding technical requirements.

Kind regards,

Razieh Sheikhpour

Academic Editor

PLOS One

Additional Editor Comments (optional):

Reviewers' comments:

Reviewer's Responses to Questions

**Comments to the Author**

Reviewer #3: All comments have been addressed

2. Is the manuscript technically sound, and do the data support the conclusions?

Reviewer #3: Yes

3. Has the statistical analysis been performed appropriately and rigorously?

Reviewer #3: Yes

4. Have the authors made all data underlying the findings in their manuscript fully available?

Reviewer #3: Yes

5. Is the manuscript presented in an intelligible fashion and written in standard English?

Reviewer #3: Yes

Reviewer #3: This is the revised version of the manuscript that was previously submitted by the authors. After thorough consideration of the changes made in this version, I have concluded that the revisions have addressed the concerns and suggestions raised during the initial review process. The improvements made enhance the overall quality and clarity of the manuscript, making it a valuable contribution to the field. Therefore, I would like to recommend this paper for publication.

**Do you want your identity to be public for this peer review?** For information about this choice, including consent withdrawal, please see our Privacy Policy

Reviewer #3: No

---

## [Editor Report · Acceptance letter]

PONE-D-25-47457R1

PLOS One

Dear Dr. Senapati,

I'm pleased to inform you that your manuscript has been deemed suitable for publication in PLOS One. Congratulations! Your manuscript is now being handed over to our production team.

Kind regards,

on behalf of

Dr. Razieh Sheikhpour

Academic Editor

PLOS One